Systematics of putative euparkeriids (Diapsida: Archosauriformes) from the Triassic of China

Sookias Roland B. 1 2 sookias.r.b@gmail.com
Sullivan Corwin 3
Liu Jun 3
Butler Richard J. 1 2
1 School of Geography, Earth and Environmental Sciences, University of Birmingham , Edgbaston, Birmingham , UK
2 GeoBio-Center, Ludwig-Maximilians-Universität München , Munich , Germany
3 Key Laboratory of Vertebrate Evolution and Human Origins of Chinese Academy of Sciences, Institute of Vertebrate Paleontology and Paleoanthropology, Chinese Academy of Sciences , Beijing , China
Anquetin Jérémy
Electronic publication date: 2014 Nov 25
Publication date: 2014
Volume: 2
Electronic Location ID: e658
Received 2014 Jun 21; Accepted 2014 Oct 17
Copyright: © 2014 Sookias et al.
Copyright year: 2014
Copyright holder: Sookias et al.
License: This is an open access article distributed under the terms of the Creative Commons Attribution License, which permits unrestricted use, distribution, reproduction and adaptation in any medium and for any purpose provided that it is properly attributed. For attribution, the original author(s), title, publication source (PeerJ) and either DOI or URL of the article must be cited.
License URL: https://creativecommons.org/licenses/by/4.0/

Keywords: Euparkeriidae, Archosauriformes, China, Triassic, Euparkeria

Funding: Emmy Noether Programme Award BU 2587/3-1 Marie Curie Career Integration Grant PCIG14-GA-2013-630123 University of Birmingham National Basic Research (973) Project 2012CB821902 Institute of Vertebrate Paleontology and Paleoanthropology of the Chinese Academy of Sciences RBS and RJB are supported by an Emmy Noether Programme Award from the Deutsche Forschungsgemeinschaft (BU 2587/3-1 to RJB), by a Marie Curie Career Integration Grant (PCIG14-GA-2013-630123 ARCHOSAUR RISE to RJB), and by the College of Life and Environmental Sciences of the University of Birmingham, UK. CS and JL are supported by the Institute of Vertebrate Paleontology and Paleoanthropology of the Chinese Academy of Sciences, Beijing, China, and JL is supported by the National Basic Research (973) Project 2012CB821902. The funders had no role in study design, data collection and analysis, decision to publish, or preparation of the manuscript.

==============================
The South African species Euparkeria capensis is of great importance for understanding the early radiation of archosauromorphs (including archosaurs) following the Permo–Triassic mass extinction, as most phylogenetic analyses place it as the sister taxon to crown group Archosauria within the clade Archosauriformes. Although a number of species from Lower–Middle Triassic deposits worldwide have been referred to the putative clade Euparkeriidae, the monophyly of Euparkeriidae is controversial and has yet to be demonstrated by quantitative phylogenetic analysis. Three Chinese taxa have been recently suggested to be euparkeriids: Halazhaisuchus qiaoensis, ‘Turfanosuchus shageduensis’, and Wangisuchus tzeyii, all three of which were collected from the Middle Triassic Ermaying Formation of northern China. Here, we reassess the taxonomy and systematics of these taxa. We regard Wangisuchus tzeyii as a nomen dubium, because the holotype is undiagnostic and there is no convincing evidence that the previously referred additional specimens represent the same taxon as the holotype. We also regard ‘Turfanosuchus shageduensis’ as a nomen dubium as we are unable to identify any diagnostic features. We refer the holotype to Archosauriformes, and more tentatively to Euparkeriidae. Halazhaisuchus qiaoensis and the holotype of ‘Turfanosuchus shageduensis’ are resolved as sister taxa in a phylogenetic analysis, and are in turn the sister taxon to Euparkeria capensis, forming a monophyletic Euparkeriidae that is the sister to Archosauria+Phytosauria. This is the first quantitative phylogenetic analysis to recover a non-monospecific, monophyletic Euparkeriidae, but euparkeriid monophyly is only weakly supported and will require additional examination. Given their similar sizes, stratigraphic positions and phylogenetic placement, the holotype of ‘Turfanosuchus shageduensis’ may represent a second individual of Halazhaisuchus qiaoensis, but no apomorphies or unique character combination can be identified to unambiguously unite the two. Our results have important implications for understanding the species richness and palaeobiogeographical distribution of early archosauriforms.

Introduction

Euparkeria capensis from the Cynognathus Assemblage Zone (Middle Triassic: Anisian) of South Africa (Ewer, 1965; Sookias & Butler, 2013) is a key species of early archosauriform that is widely regarded as approaching the ancestral archosaur body plan (e.g., Romer, 1972c; Norman & Weishampel, 1991; Parrish, 1997). Euparkeria capensis falls immediately outside of or very close to Archosauria in most phylogenetic studies (e.g., Bennett, 1996; Gower & Wilkinson, 1996; Benton, 1999; Nesbitt, 2009; Brusatte et al., 2010; Ezcurra, Lecuona & Martinelli, 2010; Nesbitt, 2011; Butler et al., 2014; Ezcurra, Scheyer & Butler, 2014), and has been used as an outgroup in numerous studies of archosaur phylogeny and morphological evolution (e.g., Perry, 1992; Carrier & Farmer, 2000; Hutchinson, 2001a; Hutchinson, 2001b; Marugán-Lobón & Buscalioni, 2003; Nesbitt, 2003; Rauhut, 2003; Seymour et al., 2004; de Ricqlès et al., 2008; Sullivan, 2010; Maidment & Barrett, 2011; Butler, Barrett & Gower, 2012; Foth & Rauhut, 2013). Several other taxa from Lower–Middle Triassic deposits around the world have historically been assigned to the group Euparkeriidae (see below; reviewed by Sookias & Butler, 2013; see also Sookias et al., in press), although no cladistic analysis has yet recovered this taxon as a monophyletic, non-monospecific entity. Most previous quantitative phylogenetic analyses of basal archosauriforms have not tested the monophyly of Euparkeriidae, because they have not included putative euparkeriid species from Poland, Russia and China (Sookias & Butler, 2013; but see Sookias et al., in press). The inclusion of these putative euparkeriid species in phylogenetic analyses has been hampered by the often fragmentary nature of their remains, and an ongoing lack of clarity with regard to their taxonomy and anatomy (Gower & Sennikov, 2000; Sookias & Butler, 2013).

Three Chinese taxa from the Anisian Ermaying Formation (see Table S2 for names in Chinese characters and Pinyin, as well as previously used romanizations) of north central China have been recently considered as putative euparkeriids worthy of further investigation (Sookias & Butler, 2013): ‘Wangisuchus tzeyii’ (Young, 1964), Halazhaisuchus qiaoensis (Wu, 1982), and ‘Turfanosuchus shageduensis’ (Wu, 1982). However, the phylogenetic relationships of these Chinese putative euparkeriids to each other, and to other archosauriforms, have never previously been tested. Given the pivotal phylogenetic position of Euparkeria capensis, testing the affinities of these taxa has the potential to clarify the relationships of major clades of early archosauriforms and patterns of character evolution during the rise of Archosauria. Here we revise the taxonomy and review the anatomy of the Chinese putative euparkeriids. We also conduct a novel phylogenetic analysis of early archosauriforms that includes two of these taxa, shedding new light on their systematic positions.

Taxonomic History of the Chinese Euparkeriids

The three species discussed here all derive from the Ermaying Formation of China and were referred to Euparkeriidae in their original descriptions. ‘Wangisuchus tzeyii’ from the upper Ermaying Formation was described by Young (1964) and referred to Euparkeriidae because of supposed similarities in the maxilla and pelvic girdle to Euparkeria capensis. ‘Wangisuchus tzeyii’ has often been subsequently considered to represent a “rauisuchian” or other pseudosuchian (i.e., a member of the ‘crocodile-line’ of Archosauria), based primarily on the presence of a suchian calcaneum within material assigned to the taxon based solely on provenance and broad morphological compatibility (e.g., Welles & Long, 1974; Krebs, 1976; Parrish, 1992; Gower, 2000; Gower & Sennikov, 2000; Borsuk-Białynicka & Sennikov, 2009; Nesbitt, 2011; Nesbitt et al., 2013). However, the species has never been adequately reassessed (Sookias & Butler, 2013) and various authors have continued to consider ‘Wangisuchus tzeyii’ a possible euparkeriid and utilize this referral in biogeographic and biostratigraphic analyses (e.g., Sennikov, 1989a; Sennikov, 1989b; Shubin & Sues, 1991; Lucas, 1998; Lucas, 2001). The species was cited as one of the earliest records of any archosaur (as a “rauisuchian”) by Benton & Donoghue (2007), and used as evidence for constraining the timing of the crocodile-bird split.

Halazhaisuchus qiaoensis and ‘Turfanosuchus shageduensis’ were described by Wu (1982) and referred to Euparkeriidae based on similarities to Euparkeria capensis, including plesiomorphies such as retention of intercentra and a “large coracoid” (Wu, 1982, p. 20). Zhen et al. (1985) considered Halazhaisuchus qiaoensis to be a “thecodont” relatively closely related to the proterosuchid Chasmatosaurus yuani, although no anatomical justification for this was given. Sennikov (1989a) and Sennikov (1989b) referred Halazhaisuchus qiaoensis, ‘Turfanosuchus shageduensis’ and ‘Wangisuchus tzeyii’ (as well as Xilousuchus sapingensis; see below) to the putative euparkeriid subgroup Dorosuchinae, along with Dorosuchus neoetus from the Middle Triassic of Russia. The basis for the referral was that these taxa were supposedly more robust than Euparkeria capensis. Parrish (1993) was apparently confusing Halazhaisuchus qiaoensis with ‘Turfanosuchus shageduensis’ when he stated that the latter was a primitive archosauriform distinct from Turfanosuchus dabanensis based on the presence of vertebral intercentra “and other features” (Parrish, 1993, p. 297), given that intercentra are present in Halazhaisuchus qiaoensis but not in ‘Turfanosuchus shageduensis’. Lucas (2001) considered both Halazhaisuchus qiaoensis and ‘Turfanosuchus shageduensis’ as euparkeriids, together with ‘Wangisuchus tzeyii’ and Euparkeria capensis (see also Lucas, 1998). Wu & Russell (2001) compared the anatomy of Halazhaisuchus qiaoensis and ‘Turfanosuchus shageduensis’ to that of Turfanosuchus dabanensis. They noted resemblances in humeral and femoral morphology between the first two species and Turfanosuchus dabanensis, but also identified differences including the presence of intercentra in Halazhaisuchus qiaoensis and discrepancies in osteoderm morphology between Halazhaisuchus qiaoensis and Turfanosuchus dabanensis. Borsuk-Białynicka & Evans (2003) tentatively supported the referral of Halazhaisuchus qiaoensis to Euparkeriidae, whereas Borsuk-Białynicka & Evans (2009) regarded the euparkeriid affinities of the taxon as doubtful.

Several other taxa from the Chinese Triassic and Lower Jurassic have historically been assigned to Euparkeriidae but are no longer regarded as potential members of the group and are not discussed in detail here. Xilousuchus sapingensis (Wu, 1981) was assigned to Euparkeriidae by Sennikov (1989a) and Sennikov (1989b), but recent analyses have reidentified it as a ctenosauriscid poposauroid (Nesbitt, Liu & Li, 2011; Butler et al., 2011; Nesbitt, 2011). Platyognathus hsui (Young, 1944) was referred to Euparkeriidae by Huene (1956), but this taxon is a crocodyliform (Wu & Sues, 1996). Turfanosuchus dabanensis (Young, 1973) was initially assigned to Euparkeriidae, but was regarded by Parrish (1993) as a suchian. The species was redescribed by Wu & Russell (2001) as a non-pseudosuchian not closely related to E. capensis, but was placed in Pseudosuchia by the most recent and extensive phylogenetic analysis of Archosauriformes (Nesbitt, 2011), and has since been identified as a member of the pseudosuchian clade Gracilisuchidae (Butler et al., 2014). ‘Fukangolepis’ barbaros (Young, 1978) was mentioned as having been referred to Euparkeriidae by Parrish (1986) but presumably this was a lapsus calami given that the holotype of the species is an indeterminate dicynodont skull fragment (Lucas & Hunt, 1993) assigned by Young (1978) to Aetosauria; the fact that Parrish (1986) cites Young (1973) for this assertion indicates Parrish may have confused ‘Fukangolepis’ barbaros with Turfanosuchus dabanensis. Finally, Yonghesuchus sangbiensis (Wu, Liu & Li, 2001) was listed without discussion as a euparkeriid by Wu & Sun (2008), but this taxon is also a gracilisuchid pseudosuchian (Butler et al., 2014).

Geological and Geographic Setting (Fig. 1)

All of the Chinese putative euparkeriid specimens discussed here are from the Ermaying Formation. The Ermaying Formation was deposited during the Triassic in a meandering fluvial environment with an east to west palaeocurrent (Liu et al., 2012). The Ermaying Formation is within the Ordos basin, and is overlain by the Tongchuan Formation (Liu, Li & Li, 2013). The specimens assigned to Halazhaisuchus qiaoensis (IVPP V6027) and ‘Turfanosuchus shageduensis’ (IVPP V6028) are from the sandstones of the lower Ermaying Formation, which is made up of yellowish pink, yellowish green, and greyish white quartz arkose (Yin, 2003). The lower Ermaying Formation has been considered early Anisian in age as a result of long-range biostratigraphic correlation with Subzone B of the Cynognathus Assemblage Zone of South Africa, based primarily on the presence of the dicynodont Kannemeyeria (Rubidge, 2005; Fröbisch, 2009). Dating of Subzone B of the Cynognathus Assemblage Zone is itself based on long-range vertebrate biostratigraphy (Hancox, 2000). Lucas (2001) argued for an Olenekian age for the lower Ermaying based on the presence of the dicynodont Shansiodon in the upper Ermaying (see below). Sues & Fraser (2010) concurred with this age assessment, based on a proposed correlation of the upper Heshanggou Formation of northern China with the lower Ermaying Formation and the presence of the typically Olenekian spore-bearing tree Pleuromeia sternbergii in the former. However, Butler et al. (2011) noted that Pleuromeia sternbergii extends into the early Anisian in Germany, and that at least part of the Heshanggou Formation may be Anisian in age. Using sensitive, high-resolution ion microprobe (SHRIMP) U-Pb dating, the age of the upper Ermaying Formation (Member II) was recently found to be 245.9 ± 3.2 Ma (Liu, Li & Li, 2013). Although the range of error encompasses the entire Anisian (currently dated as 247.2–242 Ma: Ogg, 2012; Cohen, Finney & Gibbard, 2013), this result is consistent with an Anisian age for the upper Ermaying, and by inference an early Anisian or late Olenekian date for the lower Ermaying and Heshanggou formations.

Figure 1 Localities of the putative Chinese euparkeriids.

Map of China, with inset of region, showing the localities where the holotype specimens of the three Chinese putative euparkeriid taxa reassessed in this paper were collected. White, People’s Republic of China; light grey, other countries; dark grey, ocean; thick grey lines (non-inset only), national boders; thin grey lines (grey lines in inset), province borders; thin black lines (inset only), roads; black circles (inset only), major settlements; stars, localities. Province names are in capital letters.

All material referred to ‘Wangisuchus tzeyii’ is from the white sandstones and mudstones of the upper Ermaying Formation. Rubidge (2005) and Hancox, Angielczyk & Rubidge (2013) assigned the upper Ermaying Formation to the late Anisian based on the presence of the dicynodont Shansiodon. The same genus occurs in Subzone C of the Cynognathus Assemblage Zone of South Africa (Hancox, Angielczyk & Rubidge, 2013), and the shansiodont Vinceria occurs in the Río Mendoza and Upper Puesto Viejo formations of Argentina (Hancox, 1998; Renaut & Hancox, 2001; Domnanovich & Marsicano, 2012). The proposed late Anisian date for Subzone C of the Cynognathus Assemblage Zone is itself based on long-range vertebrate biostratigraphy (Hancox, 2000). The upper Ermaying Formation was referred to the Perovkan land-vertebrate faunochron by Lucas (2010), again based upon vertebrate biostratigraphy. As noted above, new SHRIMP analyses suggest an Anisian age for the upper Ermaying Formation.

Terminology and Methods

We use the limb orientation terminology of Gower (2003), which combines that of Romer (1942) and that of Rewcastle (1980). This orientation corresponds to a fully anteriorly extended hindlimb (the anterior surfaces of hindlimb bones in descriptions of fully erect taxa such as dinosaurs thus correspond to the dorsal surfaces in our terminology), and a forelimb with the humerus fully extended posteriorly and the epipodials fully extended anteriorly (the anterior surfaces of forelimb bones in fully erect taxa thus correspond to the ventral surface of the humerus and to the dorsal surfaces of the radius and ulna here). The scapula is described with the shaft held vertically. We use the terminology of Wilson (1999) for vertebral laminae and that of Wilson et al. (2011) for vertebral fossae.

Phylogenetic analyses were carried out using the matrix of Butler et al. (2014), modified from Nesbitt (2011), with Halazhaisuchus qiaoensis and ‘Turfanosuchus shageduensis’ (not previously included by Nesbitt (2011) or Butler et al. (2014)) included in separate analyses as both distinct taxa and as a combined taxon. Additionally, we changed the scoring of osteoderm shape in Euparkeria capensis from that used by Nesbitt (2011, character 407) from “square-shaped, about equal dimensions” to “longer than wide” (see Discussion). The analyses were conducted in TNT v. 1.1 (Goloboff, Farris & Nixon, 2003; Goloboff, Farris & Nixon, 2008). We employed the same methodology as Nesbitt (2011), eliminating the same taxa from the dataset prior to analysis, with the same characters treated as ordered, and using equally weighted parsimony. An initial search using the “New Technology search” option was carried out using sectorial search, ratchet and tree-fusing options with default parameters. Minimum tree length was obtained for 1,000 separate replicates and the trees were stored in RAM. A heuristic tree search was then conducted using the stored trees, followed by TBR branch swapping. Standard bootstrap values and Bremer support values (decay indices) were calculated for each node using the inbuilt functionality of TNT and the BREMER script respectively.

Systematic Palaeontology

ARCHOSAUROMORPHA Huene, 1946 sensu Gauthier, Kluge & Rowe, 1988

ARCHOSAURIFORMES Gauthier, Kluge & Rowe, 1988 sensu Nesbitt, 2011

‘Wangisuchus’ Young, 1964

[Nomen dubium]

Type and only species. ‘Wangisuchus tzeyii’ Young, 1964.

‘Wangisuchus tzeyii’ Young, 1964

[Nomen dubium]

Holotype. IVPP V2701 (Figs. 2A–2B), an incomplete left maxilla lacking teeth.

Figure 2 Holotype and paratypes of ‘Wangisuchus tzeyii’ nomen dubium.

Holotype IVPP V2701, left maxilla, in medial (A) and lateral (B) views. Paratypes, right maxillae, in lateral views: IVPP V2702 (C); IVPP V2703 (D); IVPP V2704 (E). adm, anterodorsal margin; al, alveolus; aofo, antorbital fossa; idp, interdental plate; mas, ascending process of the maxilla; t, tooth.

Syntypes. IVPP V2702–V2704 (Figs. 2C–2E), maxillae (paratypes).

Horizon and locality. All specimens assigned to ‘Wangisuchus tzeyii’ are from the upper Ermaying Formation of Shanxi Province (Middle Triassic: Anisian). IVPP V2701 (holotype) and IVPP V2702–V2704 (paratypes) are from locality 56173, Xishiwa near Louzeyu Village, Wuxiang County (Fig. 1). This locality has been entered in the Paleobiology Database as number 101059. See Geological Setting for further information.

Remarks. The holotype maxilla, IVPP V2701 (Figs. 2A–2B), is fragmentary and undiagnostic, as are the paratype specimens. Whilst the presence of alveoli and interdental plates indicates thecodont tooth implantation (a synapomorphy of Erythrosuchus+Archosauria: Nesbitt, 2011), neither a suite of autapomorphies nor a unique combination of character states can be identified in the maxilla. The original diagnosis presented by Young (1964) was inadequate for a number of reasons: it referred to the “long and low” shape of the maxilla, but the holotype maxilla does not differ in this regard from those of most early archosauriforms; the posterior process of the maxilla was described as “pointed”, but is in fact incomplete; the anterior margin of the maxilla was described as “rounded” but is also incomplete; and teeth and other elements not preserved in the holotype were used in the diagnosis, but there is no convincing case for referring these elements to the same taxon as the holotype. We therefore consider ‘Wangisuchus tzeyii’ to be a nomen dubium. The most exclusive phylogenetic placement that can be reasonably supported for the holotype is Archosauriformes indet., based on the inferred presence of thecodont dental implantation in the maxilla. As noted above, this feature supports a position crownward of Proterosuchus (Nesbitt, 2011).

Young (1964) referred many isolated and poorly preserved postcranial elements from the type locality and other localities in the same region to ‘Wangisuchus tzeyii’, but first-hand inspection of much of this material revealed it to be undiagnostic. Furthermore, there are no compelling similarities to justify regarding even the two relatively complete paratype maxillae (IVPP V2703, V2704; Figs. 2D–2E) as necessarily conspecific with the holotype, and in fact both of these paratype maxillae appear to differ from the holotype in having a convex rather than straight anterodorsal margin (Fig. 2, adm). As discussed by several authors (Kuhn, 1976; Parrish, 1993; Gower & Sennikov, 2000; Nesbitt, 2011), an unnumbered calcaneum within this previously referred material demonstrably belongs to a suchian archosaur, but there is no evidence beyond the holotype and the calcaneum hailing from the same locality (and the generally compatible size) to support the referral of this calcaneum to ‘Wangisuchus tzeyii’.

EUPARKERIIDAE Huene, 1920 sensu Sookias & Butler, 2013

Halazhaisuchus Wu, 1982

Type and only species. Halazhaisuchus qiaoensis Wu, 1982.

Halazhaisuchus qiaoensis Wu, 1982.

Holotype. IVPP V6027 (Figs. 3–7), posterior three cervical and anterior three dorsal vertebrae in articulation with osteoderms and incomplete ribs (V6027-1), seven dorsal vertebrae in articulation with osteoderms (V6027-2), left (V6027-3) and right (V6027-4) scapulae, left (V6027-3) and partial right (V6027-4) coracoids, right humerus (V6027-5), ulna (V6027-6), and radius (V6027-7), an isolated left cervical rib (V6027-8), and an isolated median osteoderm (V6027-9). All material probably pertains to a single individual.

Figure 3 Cervical and dorsal vertebrae of Halazhaisuchus qiaoensis and cervical vertebrae of ‘Turfanosuchus shageduensis’ nomen dubium.

Posterior three cervical and anterior three dorsal vertebrae of Halazhaisuchus qiaoensis IVPP V6027-1 in right lateral (A), dorsal (B; osteoderms visible), ventral (C), anterior (D) and posterior (E) views; series of dorsal vertebrae of Halazhaisuchus qiaoensis IVPP V6027-2 in right lateral (F), dorsal (G; osteoderms visible), ventral (H), anterior (I) and posterior (J) views; cervical vertebrae of ‘Turfanosuchus shageduensis’ IVPP V6028-2 in right lateral (K; broken segments disarticulated), ventral (L), dorsal (M), anterior (N) and posterior (O) views. acpl, anterior centroparapophyseal lamina; di, diapophysis; ep, epipophysis; ic, intercentrum; ns, neural spine; ost, osteoderm; pa, parapophysis; ppdl, paradiapophyseal lamina; prdl, prezygodiapophyseal lamina; sdf, spinodiapophyseal fossa; tp, transverse process.

Figure 4 Ribs and osteoderms of Halazhaisuchus qiaoensis compared with other taxa.

Right cervical rib of Halazhaisuchus qiaoensis IVPP V6027-9 in dorsal (A) and ventral (B) views; left cervical rib (image mirrored for comparison) of Batrachotomus kupferzellensis SMNS 91046 in dorsal (C) and ventral (D) views; right paramedian osteoderm of Halazhaisuchus qiaoensis IVPP V6027-8 in dorsal (E) and ventral (F) views; right paramedian osteoderm of Euparkeria capensis UMZC T692j in dorsal (G) and ventral (H) views; paramedian osteoderms in articulation of Halazhaisuchus qiaoensis IVPP V6027-1 in dorsal (I) view (right is anterior). cap, capitulum; fl, flange; k, keel; tub, tuberculum.

Figure 5 Scapulae and coracoids of Halazhaisuchus qiaoensis and ‘Turfanosuchus shageduensis’ nomen dubium.

Left scapula and coracoid of Halazhaisuchus qiaoensis IVPP V6027-3 in lateral (A) and medial (B) views, right scapula and partial coracoid Halazhaisuchus qiaoensis IVPP V6027-4 in lateral (C) and medial (D) views, and left scapula and coracoid of ‘Turfanosuchus shageduensis’ IVPP V6028-3 in lateral (E) and medial (F) views. acr, acromion process; cof, coracoid foramen; dls, depressed lateral surface; gl, glenoid; mar, muscle attachment ridge; tu, tuber.

Figure 6 Right humeri of Halazhaisuchus qiaoensis and ‘Turfanosuchus shageduensis’ nomen dubium.

Halazhaisuchus qiaoensis IVPP V6027-5 in proximal (A), dorsal (B), lateral (C), ventral (D), distal (E) and medial (F) views, and ‘Turfanosuchus shageduensis’ IVPP V6028-4 in proximal (G), dorsal (H), lateral (I), ventral (j), distal (K) and medial (L) views. Arrows indicate dorsal direction. dpc, deltopectoral crest; ect, ectepicondyle; ectg, ectepicondylar groove; ent, entepicondyle; it, internal tuberosity; sup, supinator process.

Figure 7 Right forelimb epipodials of Halazhaisuchus qiaoensis and ‘Turfanosuchus shageduensis’ nomen dubium.

Ulna of Halazhaisuchus qiaoensis IVPP V6027-6 in proximal (A), medial (B), distal (C), lateral (D), dorsal (E), and ventral (F) views; ulna of ‘Turfanosuchus shageduensis’ IVPP V6028 in proximal (G), medial (H), distal (I), lateral (J), dorsal (K), and ventral (L) views; radius of Halazhaisuchus qiaoensis IVPP V6027-7 in proximal (M), medial (N), distal (O), lateral (P), dorsal (Q), and ventral (R) views; radius of ‘Turfanosuchus shageduensis’ IVPP V6028 in proximal (S), medial (T), distal (U), lateral (V), dorsal (W), and ventral (X) views. Arrows indicate dorsal direction. bev, bevelled surface; fos, fossa; gr, groove; ol, olecranon; ra, raised area; ri, ridge.

Horizon and locality. IVPP V6027 is from Fugu County, Shaanxi Province, China (Fig. 1). It is from the lower Ermaying Formation (Lower or Middle Triassic: late Olenekian or early Anisian), Paleobiology Database locality number 100138. See Geological Setting for further information.

Original diagnosis (paraphrased from Wu, 1982, p. 300). Relatively small pseudosuchian. Pectoral girdle well developed. Scapula exceptionally elongated and strongly expanded at both ends; ratio of scapula length to humerus length over 1.15:1; oval muscle-attachment area above glenoid with notably projecting ridge. Coracoid very large, forming two thirds of glenoid. Humerus robust, terminating in triangularly expanded apex proximally due to well-developed deltopectoral crest along proximal quarter of shaft. Radius and ulna slender, ulna with well-developed olecranon process. Vertebrae slightly amphicoelous, with elongated centra and low neural spines expanded distally; presacral vertebrae with intercentra. Cervical and anterior dorsal ribs three-headed. Row of dorsal scutes on either side of midline, scutes overlap one another and are leaf-like in outline; posterior ends of scutes grooved ventrally; in cervical and anterior dorsal regions scutes from both sides are sutured together firmly.

Revised diagnosis. Relatively small (humeral length 93.5 mm) archosauriform diagnosable by two autapomorphies: (1) strongly pronounced tuber on the scapula, for attachment of the scapular head of the m. triceps, that is circular in outline when the scapula is in lateral view, with the apex of the tuber slightly depressed (similar tubera in other taxa differ in shape, being teardrop shaped and lacking a depression, e.g., Batrachotomus kupferzellensis—Gower & Schoch, 2009); (2) pronounced muscle attachment scar on the scapula in the form of a depressed strip on the lateral surface of the blade running from anterodorsal to posteroventral, beginning at an abrupt kink in the anterior margin at around midlength of the blade. The species is further diagnosable by the following unique combination of characters: two rows of paramedian scutes with exposed surfaces at least twice as long as wide when articulated, tapering anterior processes and broad, rounded posterior margins, each having a longitudinal keel closer to the medial margin than the lateral one; large flattened flange projecting from the proximal part of the anterior margin of each cervical rib; presence of a tuber on the scapula for attachment of the scapular head of the m. triceps; presence of dorsal intercentra; epipophyses on cervical vertebrae.

The same osteoderm shape and arrangement is found in Euparkeria capensis and in some pseudosuchians (e.g., Ticinosuchus ferox—Krebs, 1965; Rauisuchus tiradentes—Lautenschlager, 2008), but differs from other stem and early crown archosaurs including proterochampsids (e.g., Chanaresuchus bonapartei—PVL 4575; single row of scutes wider than long), doswelliids (e.g., Doswellia kaltenbachi—Dilkes & Sues, 2009; multiple rows of shorter, anteriorly blunt, and more strongly sculpted scutes), and many pseudosuchians (e.g., Batrachotomus kupferzellensis—Gower & Schoch, 2009—scutes are either wider than long, lack a dorsal keel, or are blunt anteriorly; Saurosuchus galilei—Trotteyn, Desojo & Alcober 2011—exposed length of each osteoderm is shorter than the width or around the same as the width). An expanded flattened cervical rib flange is present in some crown taxa (e.g., Batrachotomus kupferzellensis—Gower & Schoch, 2009, Gracilisuchus stipanicicorum—Romer, 1972b), but is absent in most stem taxa (e.g., Euparkeria capensis—SAM-PK-5867, Proterochampsa barrionuevoi—MCZ 3408, Doswellia kaltenbachi—Dilkes & Sues, 2009). A marked scapular tuber for attachment of the m. triceps is otherwise confined to the crown and Phytosauria (e.g., Batrachotomus kupferzellensis—Gower & Schoch, 2009; Parasuchus hislopi—Nesbitt, 2011). Dorsal intercentra are absent in crown taxa (Nesbitt, 2011), differentiating Halazhaisuchus qiaoensis from pseudosuchian taxa with the same osteoderm shape (Krebs, 1965; Lautenschlager, 2008; Lautenschlager & Desojo, 2011). Pseudosuchian taxa with the same osteoderm shape as Halazhaisuchus also lack cervical epipophyses (Krebs, 1965; Lautenschlager, 2008; Lautenschlager & Desojo, 2011).

Remarks. The original differential diagnosis of Halazhaisuchus qiaoensis was insufficient because it did not adequately distinguish the taxon from other stem and early archosaurs. Many features listed (e.g., “pectoral girdle well-developed”) were not sufficiently clear or distinct to be effective in diagnosing the taxon. Other features are shared with other taxa: leaf-shaped osteoderms and presacral intercentra are shared with Euparkeria capensis (Ewer, 1965), and the vertebral features listed in the original diagnosis are essentially also shared with Euparkeria capensis (Ewer, 1965; UMZC T.692).

Description

Cervical vertebrae. IVPP V6027-1 (Figs. 3A–3E) includes what we identify as the articulated posterior three cervical vertebrae (in articulation with what we identify as the anterior three dorsals; the exact point of the cervical-dorsal transition is hard to pinpoint with certainty). The neurocentral sutures are fused. The centra of the cervical vertebrae are spool-shaped and longer than tall (as in Euparkeria capensis—SAM-PK-5867 and proterochampsids—Romer, 1972a—but contrasting with erythrosuchids—Gower, 2003—and many crown taxa—Gower & Schoch, 2009), with a low ventral keel. As in Euparkeria (SAM-PK-5867) and most early archosauriforms (e.g., Dilkes & Sues, 2009; Gower & Schoch, 2009; Romer, 1972a), in the anterior cervicals the diapophysis (Fig. 3A, di) is placed near the anterodorsal corner of the centrum, and the parapophysis (Fig. 3A, pa) is placed near the anteroventral corner; posteriorly along the column the diapophysis moves posterodorsally, the parapophysis moves dorsally to approximately halfway up the centrum, and the two become connected by a variably developed paradiapophyseal lamina (Fig. 3A, ppdl). A thick, rounded prezygadiapophyseal lamina (Fig. 3A, prdl) connects the prezygapophysis and the diapophysis as in Euparkeria capensis (SAM-PK-5867) and crown taxa (Gower & Schoch, 2009), but contrasting with some more basal taxa (e.g., Proterosuchus fergusi—NMQR 1484). A shallow spinodiapophyseal fossa (Fig. 3A, sdf) is present immediately dorsal to the diapophysis, as seen in Euparkeria capensis, but less strongly developed than in crown taxa (e.g., Gower & Schoch, 2009) and erythrosuchids (e.g., Gower, 2003). The anterior and posterior articular facets of the centra are gently concave and subcircular, as in most early archosauriforms (e.g., Ewer, 1965; Gower & Schoch, 2009; Gower, 2003; Romer, 1972a).

Some of the postzygapophyses bear epipophyses (Fig. 3A, ep), but these do not extend posteriorly beyond the postzygapophyseal articular surfaces. The presence of epipophyses contrasts with all stem archosaurs excluding Mesosuchus and Vancleavea, but is common in the crown (Nesbitt, 2011). As in many other taxa with dorsal scutes (e.g., Euparkeria capensis—SAM-PK-K8050, Batrachotomus kupferzellensis—Gower & Schoch, 2009, Jaxtasuchus salomoni—SMNS 91412), but contrasting with proterochampsids (Romer, 1972a) and taxa lacking osteoderms (e.g., erythrosuchids—Gower, 2003), the neural spines (Figs. 3A–3B, ns) widen transversely towards their distal ends to form broad, flat spine tables, each of which attains its maximum transverse width at a point slightly anterior to the midlength as in Euparkeria capensis (SAM-PK-K8050). As in Euparkeria capensis (UMZC T.692), the pre- and postzygapophyseal facets are slightly elongated ellipses in dorsal view, with the long axis of the prezygapophyseal facets running posteromedial-anterolateral, and that of the postzygapophyseal facets running posterolateral-anteromedial; the prezygapophyseal facets face anterodorsomedially and the postzygapophyseal facets face posteroventrolaterally. Unlike in most stem taxa (e.g., Erythrosuchus, Proterosuchus—Nesbitt, 2011) and some specimens of Euparkeria capensis (e.g., SAM-PK-6047A), but as in proterochampsids (e.g., Chanaresuchus barrionuevoi—SMNS 91412; Nesbitt, 2011) and crown taxa (Nesbitt, 2011), no intercentra can be identified between the cervical vertebrae, although their absence could be preservational.

Cervical ribs. IVPP V6027-1 (Figs. 3A–3E) includes three partial cervical ribs in articulation with vertebrae and IVPP V6027-8 (Figs. 4A–4B) consists of a single left cervical rib. The cervical ribs are two-headed, as in Euparkeria capensis (SAM-PK-5867) and most stem and crown archosaurs (e.g., Smilosuchus gregorii—USNM 18313, Batrachotomus kupferzellensis—Gower & Schoch, 2009, Proterosuchus fergusi—Cruickshank, 1972), and their shafts extend posteriorly, ventrally and laterally and are gently curved posteriorly, especially towards their distal ends. The tuberculum is longer than the capitulum (Fig. 4, tub, cap) and is directed medially whereas the capitulum is directed anteromedially. A dorsoventrally thin flange (Fig. 4, fl), which widens transversely as it continues proximally, extends along the anterolateral margin of each rib. A similar structure is present in several other archosauriforms, including Batrachotomus kupferzellensis (Gower & Schoch, 2009, Fig. 2M; SMNS 91046), Gracilisuchus stipanicicorum (Romer, 1972b, Fig. 7), and Smilosuchus gregorii (Nesbitt, 2011, Fig. 28J).

Dorsal vertebrae. IVPP V6027-1 (Figs. 3A–3E) includes what are probably the anteriormost three dorsal vertebrae in articulation, and IVPP V6027-2 (Figs. 3F–3J) consists of seven mid- to posterior dorsal vertebrae. The dia- and parapophyses (Figs. 3A and 3F, di, pa) are close together in the anteriormost vertebra of IVPP V6027-2, indicating that this vertebra is already a mid- or posterior dorsal. In the posteriormost vertebra of IVPP V6027-1, by contrast, the dia- and parapophyses are relatively well separated, and at least the posterior two dorsal vertebrae (what we regard here as the anteriormost dorsal may in fact be the posteriormost) preserved in this specimen can be unequivocally identified as anterior dorsals because they are in articulation with the posteriormost cervicals. Accordingly, IVPP V6027-1 and V6027-2 cannot be combined to form a continuous dorsal series.

The anterior dorsal vertebrae are generally similar to the cervical vertebrae described above, but differ in that the diapophyses are longer and dorsoventrally compressed, and are situated higher and further back on the centrum, on the suture with the neural arch. As in most stem and crown archosaurs (e.g., Euparkeria capensis—SAM-PK-5867; Batrachotomus kupferzellensis—Gower & Schoch, 2009), these differences from the cervical vertebrae become more pronounced posteriorly along the dorsal column. In successively more posterior presacral vertebrae the diapophysis and parapophysis become gradually joined, first being connected by a paradiapophyseal lamina (Figs. 3A and 3F, ppdl; already present in the more posterior cervical vertebrae) and then fusing entirely to form a single apophysis. The latter condition is present by the fourth preserved vertebra in IVPP V6027-2, although in this vertebra the parapophysis and diapophysis remain distinguishable as components of the apophysis. The diapophysis and parapophysis are indistinguishable from the fifth preserved vertebra of IVPP V6027-2 onwards; this contrasts with Euparkeria capensis, where the apophyses remain distinct units posteriorly in the column (UMZC T.692), and is more similar to the situation in Batrachotomus kupferzellensis (Gower & Schoch, 2009). A low anterior centroparapophyseal lamina (Fig. 3F, acpl) connects the parapophysis (and in more posterior vertebrae, the single fused apophysis) to the anterior margin of the centrum. A thick, rounded prezygadiapophyseal lamina (Figs. 3A and 3F, prdl) connects the prezygapophysis and the diapophysis. Very similar laminae are seen in Euparkeria capensis (UMZC T.692), but in crown taxa (e.g., Batrachotomus kupferzellensis—Gower & Schoch, 2009) and erythrosuchids (e.g., Erythrosuchus africanus—Gower, 2003) the homologous laminae are generally thinner and more sharply delimited.

A spinodiapophyseal fossa (Fig. 3F, sdf) is present dorsal to the diapophysis in the third and fifth preserved vertebrae, but the presence of this structure in other vertebrae is difficult to assess due to damage. Very similar fossae are present in Euparkeria capensis (UMZC T.692), but in erythrosuchids (Gower, 2003) and crown taxa (Gower & Schoch, 2009) these fossae are more strongly developed. The zygapophyseal facets are very similar to those of the cervical vertebrae, but the plane of articulation between the zygapophyses is more inclined than in the anteriormost two cervical vertebrae (i.e., pre- and postzygapophyseal facets face more strongly medially and laterally respectively). Intercentra (Fig. 3H, ic) are preserved in apparent articulation posterior to the fourth, fifth and sixth vertebrae of IVPP V6027-2; they are mediolaterally elongated ovals in ventral view, and their lateral tips curve dorsally which would have made them crescentic in anterior or posterior view. The intercentra appear to be more robust and larger than those recorded in Euparkeria capensis (Nesbitt, 2011), more similar to those of Erythrosuchus africanus (Gower, 2003). The dorsal ends of the neural spines (Figs. 3F and 3G, ns) are expanded into anteroposteriorly elongated oval spine tables that are covered in rugosities. These expansions are somewhat narrower than in Euparkeria capensis (UMZC T.692), being more similar in this regard to those of Jaxtasuchus salomoni (Schoch & Sues, 2013); the rugose surface contrasts with the flat surface of Euparkeria capensis (UMZC T.692) but does not approach the strongly rugose rim of Jaxtasuchus salomoni (Schoch & Sues, 2013).

Scapula. IVPP V6027-3 (Figs. 5A–5B) is a left scapula in articulation with the coracoid, and IVPP V6027-4 is a right scapula (Figs. 5C–5D). The scapula is long and bladelike, and the shaft is waisted at its dorsoventral midpoint in lateral view. This resembles the scapular form in Euparkeria capensis (SAM-PK-5867), erythrosuchids (Gower, 2003), proterochampsids (Romer, 1972a) and cursorial crown taxa (Gower & Schoch, 2009), but contrasts with the low, wide blade of Proterosuchus fergusi (Cruickshank, 1972). In posterior view the shaft of the scapula arcs in a medially concave curve. The distal margin of the scapula is damaged in both available scapulae, but is convexly curved in lateral view as preserved, with the apex of curvature closer to the posterior than to the anterior margin. The angle between the distal and posterior margins is slightly under 45° in the left scapula; the angle in the right scapula is similar, but damage prevents accurate assessment. The angle between the distal and anterior margins is slightly over 45° in the left scapula, and appears similar in the right scapula despite damage to the relevant area. These angles and the form of the distal margin appear to differ from Euparkeria capensis (SAM-PK-5867; where the apex of convexity of the distal margin is roughly central, and both anterior and posterior angles are slightly under 45°) and are more similar to Batrachotomus kupferzellensis (Gower & Schoch, 2009). The scapula possesses a pronounced posterolaterally directed tuber placed immediately dorsal to the glenoid along the posterior margin of the bone (Figs. 5A–5D, tu; the tuber on the left scapula is damaged). This tuber is for attachment of the scapular head of the m. triceps, and has a depressed lateral surface that is circular in outline in lateral view. A similar tuber is present in some “rauisuchians” (e.g., Batrachotomus kupferzellensis—Gower & Schoch, 2009; Nesbitt, 2011), but these tubera differ from that of Halazhaisuchus qiaoensis in having an elongated “teardrop” shape rather than a circular one. A distinct tuber is absent in most early archosauriforms (e.g., Euparkeria capensis—SAM-PK-5867, Erythrosuchus africanus— Gower, 2003, Smilosuchus gregorii—Nesbitt, 2011) and crown taxa, though a muscle scar is often present (Erythrosuchus africanus—NHMUK R3762a).

The acromion process (Figs. 5C–5F, acr) is larger and more prominent than in Euparkeria capensis (SAM-PK-5867), proterochampsids (Romer, 1972a), or Batrachotomus kupferzellensis (Gower & Schoch, 2009), and is not dorsally deflected as it is in erythrosuchids (Gower, 2003); its extent is similar to that of Vancleavea campi (Nesbitt, 2009), though the process is more dorsally deflected in Vancleavea campi. The lateral surface of the scapula bears a muscle attachment area (Figs. 5A and 5C, mar) in the form of a parallel ridge and groove. The groove is situated just anteroventral to the ridge, and both extend posteroventrally from a point on the anterior margin of the scapula that lies about two thirds of the way down from the dorsal end and coincides with the level at which the shaft is anteroposteriorly narrowest. On the medial surface a similarly oriented muscle attachment ridge (Figs. 5B and 5D, mar) begins on the anterior margin around two thirds of the way up from the ventral end, and terminates at the anteroposteriorly narrowest point of the shaft just anterior to the posterior margin. Neither of these ridges can be identified with certainty in other early archosauriforms (e.g., Erythrosuchus africanus—Gower, 2003; Euparkeria capensis—SAM-PK-5867, SAM-PK-6047B; Batrachotomus kupferzellensis—Gower & Schoch, 2009; Smilosuchus gregorii—Nesbitt, 2011). As in Euparkeria capensis (SAM-PK-6047B), many crown taxa (e.g., Batrachotomus kupferzellensis—Gower & Schoch, 2009; Lewisuchus admixtus—Nesbitt, 2011), and erythrosuchids (Gower, 2003), the posterior part of the shaft is substantially thicker transversely than the anterior part; the shaft is perhaps less strongly tapering in Proterosuchus fergusi (NMQR 1484). As in most early archosauriforms including Euparkeria capensis (SAM-PK-6047B), the proximal end of the shaft is strongly thickened transversely in the glenoid region, which articulates with a similarly thickened part of the coracoid. The coracoid portion of the glenoid appears to be approximately twice as large as the scapular portion, though the latter is damaged posteriorly. The scapula-coracoid suture is gently dorsally convex, with the point of maximum curvature lying around halfway along its length. The suture is clear, though the elements appear to have been firmly attached to one another, contrasting with the freely articulating elements in Euparkeria capensis (SAM-PK-5867), erythrosuchids (Gower, 2003), and Batrachotomus kupferzellensis (Gower & Schoch, 2009). Whether or not a distinct anterior notch between the elements was present (contrasting with the continuous margin in Proterosuchus fergusi, non-archosauriform archosauromorphs, and some crown taxa—Nesbitt, 2011) cannot be assessed as the anterior margin is broken.

Coracoid. IVPP V6027-3 (Figs. 5A–5B) includes a left coracoid preserved in articulation with the scapula. As in Euparkeria capensis (SAM-PK-5867, SAM-PK-6047B), erythrosuchids (Gower, 2003), proterochampsids (Romer, 1972a) and “rauisuchians” (Gower & Schoch, 2009), the coracoid is suboval with a single coracoid foramen (Figs. 5A–5B, cof) near the dorsal margin, close to the anteroposterior midpoint of the bone. The coracoid is not hooked anteriorly, unlike in phytosaurs (e.g., Smilosuchus gregorii—USNM V18313), nor does it show a postglenoid process (i.e., a notch below the glenoid), unlike in many pseudosuchians (e.g., Batrachotomus kupferzellensis—Gower & Schoch, 2009, Revueltosaurus callenderi—PEFO 34561). As in most archosauriforms (e.g., Euparkeria capensis—SAM-PK-6047B, Batrachotomus kupferzellensis—Gower & Schoch, 2009, Garjainia prima—PIN 2394), the coracoid becomes mediolaterally thicker towards its contribution to the glenoid (becoming at least five times thicker than at the anteroventral corner, where the bone is thinnest), and also immediately dorsal to the coracoid foramen. The glenoid is orientated posterolaterally, unlike the posteroventral orientation seen in some pseudosuchians and in dinosauromorphs (Nesbitt, 2011). The lateral surface of the coracoid immediately ventral to the glenoid is depressed, as in Euparkeria capensis (SAM-PK-6047B) and Batrachotomus kupferzellensis (Gower & Schoch, 2009). With the exception of this area, the lateral surface of the coracoid is smooth, with no sharp ridge from the glenoid to the anteroventral corner, unlike in Prestosuchus (UFRGS 0156-T, Nesbitt, 2011).

Humerus. IVPP V6027-5 (Figs. 6A–6F) is a right humerus. In distal view, the angle between the deltopectoral crest and the main shaft is smaller (Fig. 6E, dpc; approximately 50°) than in Euparkeria capensis (SAM-PK-5867; approximately 70°), with the crest thus protruding ventrally rather than ventrolaterally in Halazhaisuchus qiaoensis. In lateral view (Fig. 6C) the deltopectoral crest projects ventrally as a broad triangular flange and extends distally to approximately the midpoint of the shaft. This is similar to the distal extent of the crest in Euparkeria capensis (SAM-PK-5867) and Erythrosuchus africanus (Gower, 2003), but the crest does not rejoin the shaft distally as abruptly as in the latter taxon. The internal tuberosity (Fig. 6D, it) is rounded and not pronounced. As in all archosauriforms (e.g., Euparkeria capensis—SAM-PK-5867, Erythrosuchus africanus—(Gower, 2003), Batrachotomus kupferzellensis—Gower & Schoch, 2009), the humerus lacks a distinct ossified trochlea (= radial/lateral condyle) and capitellum (= ulnar/medial condyle); in ventral view the distal end is expanded, with a concave distal margin separating distally convex ect- and entepicondyles (Fig. 6D, ect, ent). The rugose and unfinished surface between these epicondyles would probably have borne a strip of cartilage connecting and covering the ect- and entepicondyles as in Caiman (see Romer, 1956, Figs. 166–167), and this cartilage might have formed a trochlea and capitellum.

The supinator process (Fig. 6B, sup) is a low, rounded ridge extending proximally along the ventrolateral edge of the shaft from the distal end; it is more pronounced than in Euparkeria capensis (SAM-PK-7700), more nearly approaching the level of development in Erythrosuchus africanus (Gower, 2003) or Batrachotomus kupferzellensis (Gower & Schoch, 2009). The distal part of the supinator process may have been more prominent in life, but the surface appears to be damaged. Dorsal to the supinator process there is a gently concave strip—this is the ectepicondylar groove (Fig. 6B, ectg), though it is less clearly developed than in Erythrosuchus africanus (Gower, 2003). It is possible that a more pronounced groove was once present distally, but is now obscured by post-mortem damage. The angle between the long axes of the distal and proximal ends of the humerus is approximately 20°; this is similar to Euparkeria capensis (SAM-PK-5867), but differs from the greater angle (>40°) seen in Prolacerta broomi (Gow, 1975).

Ulna. IVPP V6027-6 (Figs. 7A–7F) is a right ulna. The olecranon (Figs. 7A–7F, ol) is better developed than in Euparkeria capensis (SAM-PK-6047), though not as extensive as in Batrachotomus kupferzellensis (Gower & Schoch, 2009), and is rounded proximally. The proximal surface is convex dorsoventrally. The entire proximal end, including most of the olecranon, has an unfinished surface texture and was seemingly not fully ossified, contrasting with the fully ossified surface in Euparkeria capensis (SAM-PK-6047). The proximal end is suboval in proximal view, tapering dorsally and flattened medially. The shaft is slightly twisted along its length, and has the cross-sectional shape of a dorsoventrally elongated oval with a flattened medial edge. A rounded fossa midway between the dorsal and ventral edges on the medial side of the shaft, near the proximal end, in IVPP V6027-6 (Fig. 7B, fos) may be an artefact of preparation, and is not seen in other early archosauriforms (e.g., Euparkeria capensis—SAM-PK-6047, Batrachotomus kupferzellensis—Gower & Schoch, 2009, Erythrosuchus africanus—Gower, 2003).

The distal end is convex in lateral or medial view and straight in dorsal and ventral view. In distal view the distal end is a dorsoventrally elongated oval. There is a slightly raised area on the lateral surface at the proximal end of the bone (Figs. 7D–7E, ra), although, as in Euparkeria capensis (SAM-PK-6047) and other stem taxa including phytosaurs (Nesbitt, 2011), this swelling is too poorly developed to be considered a true radial tuber, contrasting with the prominent tuber present in most crown taxa (Nesbitt, 2011; Gower & Schoch, 2009). A ridge (Figs. 7D–7E, ri) extends distally along the shaft, beginning approximately 20% of the way from the proximal end and extending nearly to the distal end. Ventral and parallel to this ridge runs a groove, which becomes narrower distally. Bounding this groove ventrally is a second ridge, less well developed than the first, which angles dorsally as it extends distally. A similar ridge and groove appears to be present in Batrachotomus kupferzellensis (Gower & Schoch, 2009), but not in Erythrosuchus africanus (Gower, 2003) and Euparkeria capensis (SAM-PK-5867).

Radius. IVPP V6027-7 (Figs. 7M–7R) is a right radius. The proximal and distal ends of the shaft are formed of unfinished bone (Figs. 7M and 7O), and their outlines are mediolaterally expanded ovals. The proximal end is expanded slightly further laterally than medially, and the centre of the proximal surface is depressed, but the proximal margin is straight in dorsal view, contrasting with the slight concave curvature of this margin in Euparkeria capensis (SAM-PK-5867) and the strong concave curvature of this margin in Erythrosuchus africanus (Gower, 2003) and Batrachotomus kupferzellensis (Gower & Schoch, 2009). The ventral surface bears a groove that extends along some 80% of the length of the bone (Fig. 7R, gr), and begins and ends roughly equidistant from each end of the radius. A similar groove appears to be lacking in other early archosauriforms examined (e.g., Euparkeria capensis—SAM-PK-5867, Batrachtomus kupferzellensis—Gower & Schoch, 2009, Erythrosuchus africanus—Gower, 2003). The dorsal surface of the radius (Fig. 7Q) is flattened along about 60% of the length of the shaft, beginning near the proximal end; this flattened area is bordered both medially and laterally by an abrupt break of slope and low ridge. The radial dorsal surface appears to be slightly flattened in those early archosauriforms in which it can be observed (e.g., Erythrosuchus africanus—Gower, 2003), but not so as to form a clearly defined strip as seen in Halazhaisuchus qiaoensis. The ventral part of the distal end of the radius is slightly bevelled (Fig. 7R, bev) and rugose. The distal end is convex, as in Euparkeria capensis (SAM-PK-5867), but contrasting with the straighter margin of Batrachotomus kupferzellensis (Gower & Schoch, 2009).

Median osteoderms. IVPP V6027-1 (Figs.4A–4E, 5I) and IVPP V6027-2 (Figs. 3F–3J) include median osteoderms in articulation with cervicodorsal and dorsal vertebrae, respectively, and IVPP V6027-8 (Figs. 4E–4F) is an isolated median osteoderm. The osteoderms form two parallel rows that contact one another along the midline (Figs. 3B and 3G). The osteoderms are similar to those of Euparkeria capensis (UMZC T.692; Figs. 4G–4H) and many “rauisuchian” pseudosuchians (e.g. Ticinosuchus ferox—Krebs, 1965; Rauisuchus tiradentes—Lautenschlager, 2008) in each possessing a medially offset longitudinal keel (Figs. 4E and 4K), being more expanded laterally than medially, in being leaf shaped, and in that each osteoderm dorsally overlaps the immediately more posterior one in the same row. Each osteoderm is around twice as long anteroposteriorly as it is wide mediolaterally. Poorly defined, low rugose striations run out from the keel on the dorsal surface, similar to those seen in Euparkeria capensis (SAM-PK-13666; and not dissimilar to the ornamentation seen in the dorsal osteoderms of Batrachotomus kupferzellensis—Gower & Schoch, 2009), but contrasting markedly with the deep, pitted ornamentation in doswelliids (e.g., Jaxtasuchus—Schoch & Sues, 2013; Desojo, Ezcurra & Schultz, 2011), and no anterior smooth lamina for articulation with the preceeding osteoderm is present, contrasting with doswelliids (Desojo, Ezcurra & Schultz, 2011).

The ventral surface bears rounded rugosities at its edges, which presumably would have articulated with the similar rugosities found on the neural spines below. Similar rugosities appear to be present at the extreme edges of the osteoderms in Euparkeria capensis (Fig. 4H; SAM-PK-6047A), but are less extensive. The posterior part of the surface directly below the midline keel forms a slight depression with which the osteoderm behind would have articulated, as in Euparkeria capensis (SAM-PK-6047A). Each osteoderm overlaps the neural spines of two vertebrae (Figs. 3B and 3G), covering the anterior third of the spine of the more posterior vertebra and the posterior two thirds of the spine of the more anterior vertebra. Adjacent left and right osteoderms are level with each other anteroposteriorly rather than staggered; in Euparkeria capensis this seems to vary (SAM-PK-13666; SAM-PK-6048).

?EUPARKERIIDAE Huene, 1920 sensu Sookias & Butler, 2013

‘Turfanosuchus shageduensis’ Wu, 1982

[Nomen dubium]

Holotype and only specimen. IVPP V6028 (Figs. 3–10), mostly complete right mandible (V6028-1), six cervical vertebrae missing upper neural arches and neural spines (V6028-2), right scapula (V6028-3), coracoid (V6028-3), humerus (V6028-4), radius (V6028-7/8/9; note that the correct subnumbers for the radius, ulna and fibula are uncertain), ulna (V6028-7/8/9), femur (V6028-5), tibia (V6028-6) and fibula (V6028-7/8/9). All material probably pertains to a single individual.

Figure 8 Mandible of ‘Turfanosuchus shageduensis’ nomen dubium.

Right mandible of holotype of ‘Turfanosuchus shageduensis’, IVPP V6028-1 in lateral (A), medial (B) and dorsal (C) views. a, angular; c, coronoid; d, dentary; pra, prearticular; sa, surangular; sp, splenial; step, step between more dorsal and more ventral sections of prearticular; t, teeth.

Figure 9 Femur of ‘Turfanosuchus shageduensis’ nomen dubium.

Right femur of ‘Turfanosuchus shageduensis’ IVPP V6028-5 in dorsal (A), proximal (B), lateral (C), ventral (D), medial (E) and distal (F) views. Arrows indicate dorsal direction. ac, adductor crest; cfb, m. caudofemoralis brevis attachment; cfl, m. caudofemoralis longus attachment; ct, crista tibiofibularis; fte, m. femorotibialis externus attachment; h, head; ig, intercondylar groove; lc, lateral condyle; mc, medial condyle; ps, popliteal space; ve, ventral eminence; 4t, fourth trochanter.

Figure 10 Right hind limb epipodials of ‘Turfanosuchus shageduensis’ nomen dubium.

Tibia of ‘Turfanosuchus shageduensis’ IVPP V6028-6 in proximal (A), dorsal (B), distal (C), lateral (D), ventral (E) and medial (F) views; fibula of ‘Turfanosuchus shageduensis’ IVPP V6028 in dorsal (G), distal (H), proximal (I), lateral (J), ventral (K) and medial (L) views. Arrows indicate dorsal direction. cn, cnemial crest; lc, lateral condyle; m.if, m. iliofibularis attachment; pc, posterior condyle; ri, ridge; step, step between more medial and more lateral surfaces; ?gr, possible groove.

Horizon and locality. IVPP V6028 is from Jungar Banner, Nei Mongol Autonomous Region, China (Fig. 1), from the lower Ermaying Formation (Lower or Middle Triassic: late Olenekian or early Anisian). Paleobiology Database locality number 92436. See Geological Setting for further information.

Original diagnosis (paraphrased from Wu, 1982, p. 301). Relatively small pseudosuchian. Narrow and elongated mandible, curving dorsally anteriorly. Narrow scapula, expanded anteriorly distally. Forelimb/hind limb ratio >0.73. Humerus robust, ent- and ectepicondyles clearly defined, apex expanded into arc with large deltopectoral crest developed close to proximal end of shaft. Femur strongly sigmoid with large fourth trochanter consisting of triangular ridge. Tibia shorter than femur and more robust. Fibula slender. Vertebrae with slightly elongated centra, lacking intercentra or dorsal scutes.

Remarks. The holotype, IVPP V6028 is poorly preserved, with extensive preservational damage to the bone surface. This makes identification of details of morphology and thereby potentially diagnostic features extremely difficult. The original diagnosis of Wu (1982) was inadequate as none of the characteristics listed are autapomorphic among early archosauriforms, nor is the combination of characters unique. The mandible is described as elongate and narrow, but the ventral and dorsal parts of the ramus are damaged, making comparisons of its dorsoventral depth with other taxa uninformative. The dorsal curvature of the ramus anteriorly is common in archosauriform mandibles (e.g., Euparkeria capensis—SAM-PK-5867, Batrachotomus kupferzellensis—Gower, 1999, Erythrosuchus africanus—Gower, 2003). The degree of expansion of the scapular blade anteriorly cannot be assessed as the blade is damaged along its anterior margin. The forelimb/hind limb ratio is not unusual among primarily quadrupedal archosauriforms (e.g., 0.67 in Euparkeria capensis—Ewer, 1965). The morphology of the humeral condyles and deltopectoral crest is not unusual, being similar to that of Euparkeria capensis (SAM-PK-5867) and Batrachotomus kupferzellensis (Gower & Schoch, 2009). The sigmoid shape and degree of projection of the fourth trochanter of the femur is very similar to that of Euparkeria capensis (SAM-PK-5867) and Dorosuchus neoetus (Sookias et al., in press), and also to proterochampsids (e.g., Chanaresuchus bonapartei—MCZ 4036) and some crown taxa (e.g., Batrachotomus kupferzellensis—Gower & Schoch, 2009). The tibia is neither unusually short nor unusually robust (cf. Euparkeria capensis—SAM-PK-6047B, Dorosuchus neoetus—Sookias et al., in press), and similarly slender fibulae are found in other archosauriform taxa (e.g., Gracilisuchus stipanicicorum—PVL 4597). The dimensions of the vertebral centra and the presence or absence of dorsal osteoderms are uncertain as the dorsal portions of the vertebrae are lacking due to damage. Intercentra may have simply not been preserved (the centra are bevelled, indicating they may have been present), or only have been present elsewhere in the column.

IVPP V6028 was designated by Wu (1982) as the holotype of a putative new species of the genus Turfanosuchus, ‘T. shageduensis’. The type species of Turfanosuchus, Turfanosuchus dabanensis, is from the Kelamayi Formation (Middle Triassic) of Xinjiang, China. Subsequently Gower & Sennikov (2000) expressed doubts that ‘Turfanosuchus shageduensis’ and Turfanosuchus dabanensis were congeneric, and noted instead the strong similarities of ‘Turfanosuchus shageduensis’ to Halazhaisuchus qiaoensis from the same formation. However, we find ‘Turfanosuchus shageduensis’ to be too incomplete and poorly preserved to refer to either of these genera.

The most exclusive phylogenetic placement that we find to be unequivocally supported for ‘Turfanosuchus shageduensis’ is Archosauriformes indet. based on thecodont implantation in the mandible, supporting a placement within Archosauriformes crownward of Proterosuchus (Nesbitt, 2011). ‘Turfanosuchus shageduensis’ has a femur that is similar in its sigmoid shape and possession of a symmetrical and rounded fourth trochanter to those of proterochampsids (e.g., Chanaresuchus bonapartei—MCZ 4036), Dorosuchus neoetus (Sookias et al., in press), and some crown archosaurs (e.g., Batrachotomus kupferzellensis—Gower & Schoch, 2009), indicating that it can probably be placed higher on the stem than erythrosuchids (Gower, 2003) or Proterosuchus (Nesbitt, 2011). We would suggest that tentative assignment to Euparkeriidae is reasonable, based on the lack of a groove on the proximal surface of the femur (contrasting with proterochampsids, e.g., Chanaresuchus bonapartei—MCZ 4035) and apparent weak development of vertebral fossae, unlike the stronger development in crown taxa (e.g., Batrachotomus kupferzellensis—Gower & Schoch, 2009). However, the vertebrae are poorly preserved, and no autapomorphy unites ‘Turfanosuchus shageduensis’ with Euparkeria capensis or other putative euparkeriids. Whilst the preserved morphology does not preclude the holotype of ‘Turfanosuchus shageduensis’ from being the same taxon as Halazhaisuchus qiaoensis, and the size and stratigraphic position are broadly comparable, the two share no unambiguous synapomorphies.

Description

Mandible. IVPP V6028-1 (Fig. 8; measurements for this and all other elements given in Table S2) is a poorly preserved right mandibular ramus lacking the posteriormost part. Extensive cracking and damage to the external surfaces of most elements prevents accurate identification of sutures. The mandible is ventrally convex in lateral view. The ramus is long anteroposteriorly and shallow dorsoventrally, but the heavily damaged and compressed posterior end of the ramus was probably deeper in life. A mandibular fenestra cannot be identified with certainty due to poor preservation. At least five teeth (Fig. 8, t) and three additional empty alveoli can be identified, and the dentary appears to be long enough to accommodate approximately 12 teeth in total, but the exact posterior extent of the dentary is unclear. This is a similar number to Euparkeria capensis (nine visible and space for at least 11 in SAM-PK-5867), Batrachotomus kupferzellensis (Gower, 1999) and Erythrosuchus africanus (approximately 13—Gower, 2003), but fewer than in long-snouted taxa such as Doswellia kaltenbachi (approximately 35—Dilkes & Sues, 2009) or Chanaresuchus bonapartei (approximately 18—Romer, 1971). The teeth are close to circular in cross-section, seemingly contrasting with the mediolaterally flattened teeth of Euparkeria capensis (SAM-PK-5867), but further details of their morphology cannot be discerned. The prearticular (Fig. 8, pra) can be identified posteriorly on the medial side, expanding in dorsoventral depth towards its posterior end. The prearticular is mediolaterally thin and dorsoventrally deep with an almost flat (very slightly medially convex in posterior view) and smooth medial surface, as seen in Euparkeria capensis (SAM-PK-5867) or Erythrosuchus africanus (Gower, 2003). An abrupt, approximately longitudinal step (Fig. 8, step) demarcates a slightly inset ventral portion of the medial surface of the prearticular that would have been covered by the angular in the intact mandible.

Contributing to the anterior portion of the ramus are fragments of bone that, based on their positions, probably represent parts of the splenial (Fig. 8, sp) and coronoid (Fig. 8, c); the portion of the ramus formed by these elements is medially convex in posterior view. The possible coronoid medial to the tooth row is transversely wider in dorsal view than is the part of the dentary lateral to the tooth row. The ventrolateral edge of the dentary (Fig. 8, d) is convex in anterior view, as in most other early archosauriforms (e.g., Euparkeria capensis—SAM-PK-5867, Erythrosuchus africanus—BPI 5207). Ventrally, the dentary and splenial (Fig. 8, sp) are separated by a narrow gap, but this may be due to post-mortem damage, with no such gap present in Euparkeria capensis (SAM-PK-5867). The dorsolateral edge of the area of the mandibular ramus that is likely formed by the surangular (Fig. 8, sa) is convex in anterior view, and was clearly dorsally convex in lateral view when intact. The area of the mandibular ramus that is likely formed by the angular (Fig. 8, a) forms the ventralmost point of the jaw, as in most early archosauriforms (e.g., Euparkeria capensis—SAM-PK-5867, Erythrosuchus africanus—Gower, 2003, Batrachotomus kupferzellensis—Gower & Schoch, 2009, Proterosuchus fergusi—Cruickshank, 1972). The lateral surface of the angular is dorsoventrally convex, and the angular tapers posteriorly in lateral view.

Cervical vertebrae. IVPP V6028-2 (Figs. 3K–3O) consists of six very poorly preserved, articulated cervical vertebrae, all of which lack the dorsal part of the neural arch including the neural spine. As in Halazhaisuchus qiaoensis and Euparkeria capensis (SAM-PK-5867—see above), the centra of the cervical vertebrae are spool-shaped and longer than tall as preserved, with low ventral keels. In the anterior of the preserved column the parapophysis (Fig. 3K, pa) is placed anteroventrally and the diapophysis (Fig. 3K, di) is directly dorsal to it, separated by a narrow gap; this differs from the anteriormost preserved cervical of Halazhaisuchus, where the apophyses are broadly separated, though the difference may in part reflect dorsoventral compression in ‘Turfanosuchus shageduensis’. Posteriorly in the column the diapophysis moves slightly dorsally, but no paradiapophyseal lamina is visible, unlike in Halazhaisuchus qiaoensis. As in Halazhaisuchus qiaoensis (see above), crown taxa and proterochampsids, but contrasting with most stem taxa (Nesbitt, 2011), no intercentra can be identified; given that the vertebrae are bevelled, they may have been lost post-mortem as part of the preservation process. The vertebrae of ‘Turfanosuchus shagduensis’ are slightly longer and lower in their proportions than those of Halazhaisuchus qiaoensis, but this may reflect post-mortem compression of the former given that their ventral surfaces are flattened.

Scapula. IVPP V6028-3 is a poorly preserved right scapula in articulation with a partial coracoid (Figs. 5E–5F). The scapula is broadly similar to that of Halazhaisuchus qiaoensis, with a long, wasted and bladelike shaft, arcing medially. As in Euparkeria capensis (Ewer, 1965; SAM-PK-5867) and Halazhaisuchus qiaoensis, the shaft is wider mediolaterally at its posterior margin, and the proximal end is strongly thickened at the glenoid. The margin of the bone is broken in the region in which the tuber for the m. triceps would have been placed, but there is a swelling in this position that probably represents what remains of some form of tuber following post-mortem damage (contrasting with Euparkeria capensis, which lacks such a tuber—SAM-PK-5867), though whether it was similar in form to the circular tuber of Halazhaisuchus qiaoensis or the teardrop shaped tubera of “rauisuchians” (e.g., Batrachotomus kupferzellensis—Gower & Schoch, 2009; Nesbitt, 2011) cannot be ascertained. The muscle attachment ridges identified in Halazhaisuchus qiaoensis are not visible, though they may have been obliterated due to the poor preservation of the surface of the scapula. The scapula has a mediolaterally thinner and slightly anteroposteriorly wider shaft than either scapula of Halazhaisuchus qiaoensis, though this may be in part due to post-mortem mediolateral compression. The scapula and coracoid have broken apart, and damage to their articulating surfaces means that the nature of their articulation is uncertain, though the suture appears to have been roughly horizontal. The scapular contribution to the glenoid is approximately 80% that of the coracoid.

Coracoid. IVPP V6028-3 (Figs. 5E–5F) includes a partial right coracoid in articulation with the scapula, though the two have broken apart post mortem. The coracoid is broadly similar to that of Halazhaisuchus qiaoensis (see above) in being suboval, thickening towards the glenoid, and showing a lateral depression ventral to the glenoid. A coracoid foramen cannot be identified, though this may be due to preservation. The anterior of the coracoid is lost, and thus whether it was hooked anteriorly (unlike in Halazhaisuchus qiaoensis or Euparkeria capensis—SAM-PK-5867, but as in phytosaurs—e.g., Smilosuchus gregorii, USNM V18313) cannot be ascertained. As in Halazhaisuchus qiaoensis (and unlike in some pseudosuchians and dinosauromorphs—Nesbitt, 2011), the glenoid is orientated posterolaterally. The lateral surface is seemingly smooth, as in Halazhaisuchus qiaoensis, though not well preserved.

Humerus. IVPP V6028-4 (Figs. 6G–6L) is a right humerus, broadly similar in morphology to that of Halazhaisuchus qiaoensis. The deltopectoral crest is broken, but appears to be more laterally directed than in Halazhaisuchus qiaoensis and more similar to that of Batrachotomus kupferzellensis (Gower & Schoch, 2009); this may however be due to mediolateral compression of the entire proximal end of the humerus in Halazhaisuchus qiaoensis, as evidenced by extensive cracks across the surface of the bone. The position of the crest on the humeral shaft does not differ noticeably from that seen in Euparkeria capensis (SAM-PK-5867), contra Wu (1982); like that of Halazhaisuchus qiaoensis, the deltopectoral crest does not rejoin the shaft as abruptly distally as in Erythrosuchus africanus (Gower, 2003). The internal tuberosity (Figs. 6G–6L, it) is visible as a rounded medial projection from near the proximal margin in ventral view. The internal tuberosity is more prominent than that of Halazhaisuchus qiaoensis, and is more similar in extent to that of Euparkeria capensis (SAM-PK-7700; though the tuberosity is not so clearly delimited by a proximal groove/depression in Euparkeria capensis as in ‘Turfanosuchus shageduensis’), though this may again partly reflect mediolateral compression of the proximal end in Halazhaisuchus qioaensis. The distal part of the supinator process is damaged, as in Halazhaisuchus qiaoensis.

Ulna. IVPP V6028 includes a right ulna (Figs. 7G–7L) that is either IVPP V6028-7, IVPP 6028-8 or IVPP V6028-9 (it is unclear which of these numbers refers to the ulna of IVPP V6028, and which ones to the radius and fibula). The ulna is broadly similar to that of Halazhaisuchus qiaoenis, and details of its morphology are difficult to assess reliably due to extensive damage. However, the olecranon appears to be longer, more tapered, and more dorsally directed than in Halazhaisuchus qiaoensis; the olecranon is similar in extent to that of Batrachotomus kupferzellensis (Gower & Schoch, 2009), but this structure is less rounded in ‘Turfanosuchus shageduensis’. The distal end of the ulna also appears to be more expanded dorsally than in Halazhaisuchus qiaoensis, forming a separate lobe proximodorsal to the ventrodistal margin; a similar lobe is seemingly absent in other early archosauriforms (e.g., Batrachotomus kupferzellensis—Gower & Schoch, 2009, Erythrosuchus africanus—Gower, 2003), and this feature may be an artefact. No details of the surface of the shaft (ridges, grooves) are discernible due to preservation. No proximal fossa on the medial surface similar to that of Halazhaisuchus qiaoensis (which may be an artefact—see above) is visible.

Radius. IVPP V6028 includes a poorly preserved right radius (Figs. 7S–7X; either IVPP V6028-7, IVPP V6028–8, or IVPP V6028-9, see above). The radius is generally similar to that of Halazhaisuchus qiaoensis. Like in the ulna, extensive damage prevents details of morphology such as ridges and grooves from being discerned. The radius appears to be slightly more slender than that of Halazhaisuchus qiaoensis, especially distally, and to lack any lateral expansion proximally (unlike in Halazhaisuchus qiaoensis where the proximal end is equally expanded laterally and medially), though the latter feature may be due to preservation as this does not occur in other early archosauriforms (e.g., Batrachotomus kupferzellensis—Gower & Schoch, 2009, Erythrosuchus africanus—Gower, 2003).

Femur. IVPP V6028-5 (Fig. 9) is a right femur. The shaft is sigmoidal, as in Euparkeria capensis (SAM-PK-5867), proterochampsids (Romer, 1972a), and many crown taxa (Gower & Schoch, 2009), but contrasting with the straighter shaft of erythrosuchids (Gower, 2003) and proterosuchids (Cruickshank, 1972). In distal view, the angle of offset between the long axes of the distal and proximal ends (40–50°) is similar to the corresponding angle in Euparkeria capensis (SAM-PK-6047B), but contrasts strongly with the 60°–90°deflection seen in Vancleavea campi (Nesbitt, 2009) and phytosaurs (Nesbitt, 2011) and with the <20° deflection in Jaxtasuchus salomoni (Schoch & Sues, 2013). The proximal end of the femur is a dorsomedially-ventrolaterally elongated oval in proximal view (Fig. 9B); the bone surface is rugose and very slightly concave, indicating the presence of a cartilaginous proximal end in life, but is not pronouncedly depressed as in erythrosuchids (Gower, 2003) and proterochampsids (Chanaresuchus bonapartei—PVL 6244). A low ridge (= medial tuber of Nesbitt, 2011) extends distally along the ventral surface of the femur, beginning at the proximal margin then nearly merging indistinguishably with the bone surface, before redeveloping into a clear fourth trochanter (Figs. 9D–9F, 4t). As in Euparkeria capensis (SAM-PK-5883, SAM-PK-6047B) and most non-crown taxa (e.g., Chanaresuchus bonapartei—MCZ 4035, Dorosuchus neoetus—Sookias et al., in press, phytosaurs—Nesbitt, 2011) there is thus only a single medial tuber, rather than an anteromedial and posteromedial tuber. As in Euparkeria capensis (SAM-PK-5883, SAM-PK-6047B), proterochampids (Romer, 1972a) and crown taxa (Gower & Schoch, 2009), the fourth trochanter forms a laterally convex arc in ventral view. The apex of the trochanter is halfway between the proximal and distal ends of this structure and situated closer to the medial margin of the femur than to the lateral margin; the trochanter is mediolaterally widest at this point. The presence of a symmetrical fourth trochanter contrasts with the large, blade-like, asymmetric attachment ridge for the m. caudifemoralis seen in erythrosuchids (Gower, 2003), proterosuchids (Cruickshank, 1972), and non-archosauriform archosauromorphs (e.g., Trilophosaurus buettneri—Spielmann et al., 2008), and is far more similar to taxa just outside the crown including Euparkeria capensis (SAM-PK-5883, SAM-PK-6047B), Dorosuchus neoetus (Sookias et al., in press) and proterochampsids (Romer, 1972a), and to crown taxa (e.g., Batrachotomus kupferzellensis—Gower & Schoch, 2009).

A raised ring of bone surrounding a rugose depression lateral to the proximal end of the trochanter (Fig. 9E, cfb) may, following the arrangement in Alligator (Romer, 1923; Hutchinson, 2001b; Schachner, Manning & Dodson, 2011), be the area of insertion for the m. caudofemoralis brevis, and the trochanter itself in addition to a proximomedially adjacent rugose area (Fig. 9E, cfl) may represent the area of insertion for the m. caudofemoralis longus (again following Alligator—Romer, 1923; Hutchinson, 2001b; Schachner, Manning & Dodson, 2011). Similar areas of probable muscle scarring are visible in Euparkeria capensis (SAM-PK-5883), though they are not readily identifiable even in some taxa otherwise similar in femoral morphology (e.g., Batrachotomus kupferzellensis—Gower & Schoch, 2009). A rounded and raised area on the lateral surface of the femur (Fig. 9B, fte), about one third of the shaft length from the proximal end, may mark the proximal part of the area of origin of the m. femorotibialis externus (as identified in Alligator—Romer, 1923; Hutchinson, 2001b; Schachner, Manning & Dodson, 2011). This raised area is adjacent to a slight bulge on the ventrolateral margin of the femur, referred to here as the ventral eminence (Fig. 9B, ve). Whilst the ventral eminence is clearly present in Euparkeria capensis (SAM-PK-5883, SAM-PK-6047B) and Erythrosuchus africanus (Gower, 2003), the m. femorotibialis externus area is not readily identifiable in these taxa.

The shaft has an egg-shaped cross-section, in that the ventral margin of the shaft is narrower mediolaterally than the dorsal margin and narrows further to form the adductor crest (Fig. 9D, ac) as it passes distally. The distal end of the femur is divided into lateral and medial condyles (Fig. 9E, lc, mc) that are separated by an intercondylar groove distally (Fig. 9F, ig) and dorsally, and by a shallowly depressed popliteal space ventrally (Fig. 9D, ps). This arrangement is similar to that in Euparkeria capensis (SAM-PK-5883, SAM-PK-6047B), though the condyles are less strongly defined in the latter taxon, and is also found in crown taxa (Gower & Schoch, 2009), proterochampsids (Romer, 1972a), doswelliids (Schoch & Sues, 2013), and Dorosuchus neoetus (Sookias et al., in press). In erythrosuchids, the condyles are much less strongly defined, though they project further beyond the midshaft width (Gower, 2003). The lateral condyle bears a tapered, ventrally projecting crista tibiofibularis (Fig. 9C, ct); damage to the lateral margin of the condyle and crista tibiofibularis impedes assessment of whether the angle between the two was obtuse (as in Euparkeria capensis and most stem and avian-line archosaurs) or close to a right angle (as in some crown pseudosuchians including Batrachotomus kupferzellensis) (see Nesbitt, 2011). The bone surface of the distal end (Fig. 9F) is rugose, indicating a large cartilaginous epiphysis in life.

Tibia. IVPP V6028-6 (Figs. 10A–10F) is a right tibia. The proximal end of the tibia is around twice as expanded dorsoventrally (23 mm versus 13.2 mm as preserved) and mediolaterally (21.5 mm versus 14.7 mm as preserved) as the distal end; this is greater than the degree of expansion seen in Euparkeria capensis (SAM-PK-5867) or Batrachotomus kupferzellensis (Gower & Schoch, 2009), and is more similar to the expansion seen in proterochampsids (Romer, 1972a), erythrosuchids (Garjainia prima—PIN 951/28), and Dorosuchus neoetus (Sookias et al., in press, though the expansion is less symmetrical in this taxon). The proximal end has relatively straight dorsomedial, dorsolateral and ventrolateral edges and a convexly curved ventromedial edge in proximal view (Fig. 10A). The dorsal margin of the proximal end is expanded to form a small cnemial crest (Figs. 10A and 10B, cn), similar to the small bump in Euparkeria capensis (SAM-PK-5867) and contrasting with the more pronounced crest of dinosauromorphs (Nesbitt, 2011). The ventrolateral corner of the proximal end is very slightly expanded to form an indistinct posterior condyle (Fig. 10A, pc), as in Euparkeria capensis (SAM-PK-5867).

As in most early archosauriforms (but contrasting with theropods—Nesbitt, 2011), the proximal surface of the tibia is convex overall; however, it is interrupted by a dorsoventrally elongated concavity that is closer to the lateral margin of the proximal surface than the medial margin. The shaft of the tibia displays a dorsally convex curvature in lateral view (Fig. 10D), contrasting with the equally dorsally and ventrally concave shaft of Euparkeria capensis (SAM-PK-5867) and more similar to the shape in Dorosuchus neoetus (Sookias et al., in press). The cross-sectional shape of the shaft is a mediolaterally compressed ellipse, less close to circular than in Euparkeria capensis (SAM-PK-6047B) and more similar to Dorosuchus neoetus (Sookias et al., in press), though this may in part be due to damage. As preserved, the distal end of the tibia has the outline of an oval elongated along a ventrolateral-to-dorsomedial axis (Fig. 10C), and is very slightly concave. This differs markedly from the almost circular distal end in Euparkeria capensis (SAM-PK-6047B), and the elliptical but less strongly elongated one in Dorosuchus neoetus (Sookias et al., in press), though this may again in part be due to post mortem damage. No definite attachment site for the m. puboischiotibialis can be identified (unlike the condition in Erythrosuchus africanus, Gower, 2003). There is a step (Fig. 10F, step) on the medial surface of the tibia, beginning around one quarter of the way down the shaft. This step separates the more prominent ventral part of the medial surface of the tibia from the more subdued dorsal part. No similar feature can be identified in other early archosauriforms (e.g., Euparkeria capensis—SAM-PK-6047B, Dorosuchus neoetus—Sookias et al., in press, Erythrosuchus africanus—Gower, 2003), and it may be a preservational artefact.

Fibula. IVPP V6028 includes a right fibula (Figs. 10G–10L; either IVPP V6028-7, IVPP V6028–8, or IVPP V6028-9, see above). The fibula is long and slender (ratio of shaft diameter to shaft length is lower than in Euparkeria capensis—SAM-PK-5867—and more similar to Gracilisuchus stipanicicorum—Lecuona & Desojo, 2011), relatively straight (showing some curvature, as in Euparkeria capensis—SAM-PK-5867 and Gracilisuchus stipanicicorum—Lecuona & Desojo, 2011), but contrasting with the curved fibula of, e.g., Postosuchus kirkpatricki—TTU-P 9002, and the straight element of dinosauromorphs—Nesbitt, 2011) and flattened mediolaterally. The proximal end of the fibula is missing, but the proximalmost preserved part of the bone bears an eminence on the lateral surface (Fig. 10J, m.if) that was interpreted by Wu (1982) as the insertion site for the m. iliofibularis (corresponding to the anterior trochanter of e.g., Borsuk-Białynicka & Sennikov, 2009). This interpretation is plausible, but the attachment would then be more proximally positioned than in most stem and early archosaurs (e.g., Nesbitt, 2011: Fig. 41). A possible exception is Osmolskina (Borsuk-Białynicka & Sennikov, 2009), but no fibula has been assigned to this taxon with more than tentative certainty. However, a proximally placed m. iliofibularis insertion is characteristic of derived pseudosuchians (e.g., Crocodylus niloticus: Borsuk-Białynicka & Sennikov, 2009).

The shaft tapers mediolaterally and dorsoventrally for more than half of its preserved length before reexpanding distally. The long axes of the distal part of the shaft and the proximalmost preserved part are offset by around 75°. The shaft is oval in cross-section, but the dorsal surface is pinched to form an elongated ridge (Fig. 10G, ri). The distal end of the shaft is strongly expanded ventrolaterally to dorsomedially, and the ventrolateral margin of the distal end is much wider in distal view than the dorsomedial margin. Euparkeria capensis (SAM-PK-5867) lacks such a strong ventrolateral-dorsomedial expansion but some degree of expansion is seen in most early archosauriforms (e.g., Postosuchus kirkpatricki—TTU-P 9002, Proterosuchus fergusi—NM QR 880). A small groove (Fig. 10K, ?gr) runs proximodistally along the ventral surface of the fibula near the distal end, though this may be an artefact of poor preservation as it is not shared with other taxa (e.g., Postosuchus kirkpatricki—TTU-P 9002, Proterosuchus fergusi—NM QR 880, Batrachotomus kupferzellensis—Gower & Schoch, 2009). In lateral view, the distal margin of the fibula is embayed between dorsal and ventral rounded convexities, and the ventral end of the distal margin is more proximal than the dorsal end, probably in order to form an articulation for the astragalus, as seen in many pseudosuchians (e.g., Postosuchus kirkpatricki—TTU-P 9002, Hesperosuchus agilis—AMNH FR 6758), but unlike in Euparkeria capensis (SAM-PK-5867) or Proterosuchus fergusi (NM QR 880). The lateral surface of the distal end is depressed at its dorsoventral midpoint.

Phylogenetic Relationships ofHalazhaisuchus Qiaoensis and ‘Turfanosuchus Shageduensis’

Although we find ‘Turfanosuchus shageduensis’ to be a nomen dubium due to lack of diagnostic characters, we placed both it and Halazhaisuchus qiaoensis in a phylogenetic analysis, as both taxa could be scored for a reasonable number of characters. Our initial phylogenetic analysis (Fig. 11) including Halazhaisuchus qiaoensis and ‘Turfanosuchus shageduensis’ yielded 810 most parsimonious trees (MPTs) of 1257 steps with a consistency index (CI) of 0.384 and a retention index (RI) of 0.793. Halazhaisuchus qiaoensis and ‘Turfanosuchus shageduensis’ were found to be sister taxa, forming a clade that was in turn placed as sister to Euparkeria capensis. This result is consistent with the general similarity of ‘Turfanosuchus shageduensis’ to Halazhaisuchus qiaoensis, but the taxa are not united by any autapomorphies. The result supports a monophyletic Euparkeriidae, consisting of Euparkeria capensis, Halazhaisuchus qiaoensis, and potentially ‘Turfanosuchus shageduensis’, that forms the sister clade to Archosauria+Phytosauria. However, this Euparkeriidae is supported only by one local apomorphy—character 407, presacral osteoderms that are longer than wide—and osteoderms are not preserved in ‘Turfanosuchus shageduensis’. The sister grouping of Halazhaisuchus qiaoensis and ‘Turfanosuchus shageduensis’ is supported by a single local apomorphy: 219, teardrop-shaped tuber on posterior edge of scapula present (following the wording of (Nesbitt, 2011)—the tuber of Halazhaisuchus qiaoensis is in fact circular, but is almost certainly homologous with the teardrop shaped tubera of other taxa). Bootstrap support for the node Euparkeriidae+(Archosauria+Phytosauria) is 60%, with a Bremer support of three, but bootstrap support for Euparkeriidae and for Halazhaisuchus qiaoensis+‘Turfanosuchus shageduensis’ is <40% and Bremer support for both nodes is one. Seven extra steps were required to find a monophyletic clade composed of ‘Turfanosuchus shageduensis’, Halazhaisuchus qiaoensis and Turfanosuchus dabanensis (whether or not ‘Turfanosuchus shageduensis’ and Halazhaisuchus qiaoensis were constrained to be sister taxa). Turfanosuchus dabanensis was placed as the sister taxon of Gracilisuchus+Yonghesuchus within Pseudosuchia, as found by Butler et al. (2014). Nineteen extra steps were required to recover a monophyletic Euparkeriidae composed of a combined Halazhaisuchus qiaoensis, ‘Turfanosuchus shageduensis’, Turfanosuchus dabanensis and Euparkeria capensis (whether or not ‘Turfanosuchus shageduensis’ and Halazhaisuchus qiaoensis were constrained to be sister taxa). One extra step was required to place a ‘Turfanosuchus shageduensis’ and Halazhaisuchus qiaoensis clade one node outside Euparkeria+(Phytosauria+Archosauria), or to place such a clade one node above Euparkeria as the sister taxon to Phytosauria+Archosauria.

Figure 11 Phylogenetic position of Halazhaisuchus qiaoensis and ‘Turfanosuchus shageduensis’ nomen dubium.

Strict consensus of 810 most parsimonious trees of length 1257 steps, showing the phylogenetic positions of Halazhaisuchus qiaoensis and ‘Turfanosuchus shageduensis’. Consistency index = 0.384; retention index = 0.793. Numbers below nodes are bootstrap values (before the slash) and decay indices (after the slash) for the nodes in question.

As the possibility cannot be excluded that ‘Turfanosuchus shageduensis’ represents a second specimen of Halazhaisuchus qiaoensis, we also conducted analyses with the taxa treated as a single operational taxonomic unit (OTU). The recovered tree topology (excepting of course the sister group relationship of the two taxa in question) and character optimizations were identical to those found in the previous analysis when Halazhaisuchus qiaoensis and ‘Turfanosuchus shageduensis’ were combined as a single taxon, and support values differed only slightly (Bremer support of four for Archosauria+Phytosauria). This analysis recovered 270 MPTs of 1276 steps with a CI of 0.379 and an RI of 0.787. Turfanosuchus dabanensis was placed as the sister taxon of Gracilisuchus+Yonghesuchus within Pseudosuchia, as found by Butler et al. (2014). Seven extra steps were required to place Turfanosuchus dabanensis as the sister taxon to the combined Halazhaisuchus qiaoensis. Nineteen extra steps were required to recover a monophyletic Euparkeriidae composed of a combined Halazhaisuchus qiaoensis OTU, Turfanosuchus dabanensis and Euparkeria capensis.

Discussion

We consider ‘Wangisuchus tzeyii’ to be a nomen dubium due to the undiagnostic nature of the holotype material. Whilst some of the material currently assigned to the taxon may indeed pertain to a euparkeriid or euparkeriid-grade species, the specimens are too fragmentary and poorly preserved for a reasonable assessment of their systematic position to be made. The problem is compounded by the lack of convincing evidence that any of the different specimens pertain to the same individual or taxon, especially given that other archosauromorphs (e.g., Shansisuchus shansisuchus) were collected from the same localities and strata.

Although Halazhaisuchus qiaoensis and the holotype of ‘Turfanosuchus shageduensis’ were found to be sister taxa, and there are no major differences in morphology between overlapping elements of the two taxa, we feel the morphology of ‘Turfanosuchus shageduensis’ is not well enough preserved to allow synonymization of the taxa. The diagnostic characters of Halazhaisuchus qiaoensis are not preserved (and may not have been present) in ‘Turfanosuchus shageduensis’, and we regard ‘Turfanosuchus shageduensis’ as only diagnosable as an indeterminate archosauriform, and possibly a euparkeriid. ‘Turfanosuchus shageduensis’ shares all characters for which it could be scored with Euparkeria capensis and Halazhaisuchus qiaoensis, except for the presence on the scapula of a tuber for the m. triceps, which it only shares with the latter. Thus tentative placement of ‘Turfanosuchus shageduensis’as a euparkeriid appears reasonable, but the only potential euparkeriid autapomorphy identified here—osteoderm shape—is not preserved in the taxon. Even if ‘Turfanosuchus shageduensis’ is not a euparkeriid, it can be safely concluded that it is a stem archosaur of a similar grade to Euparkeria capensis and Halazhaisuchus qiaoensis.

Our phylogenetic analysis constitutes only the third test of the existence of a monophyletic, non-monospecific Euparkeriidae, the first being an analysis by Ezcurra, Lecuona & Martinelli (2010) that included the putative euparkeriids Osmolskina czatkowicensis and Euparkeria capensis, and the second being an analysis by Sookias et al. (in press) that included the putative euparkeriids Dorosuchus neoetus and Euparkeria capensis. As neither of these previous studies found the putative euparkeriids included in the analysis to be sister taxa, our analysis is the first to recover a monophyletic, non-monospecific euparkeriid clade. Our ongoing work is focused on developing a more extensive dataset to simultaneously test the positions of Euparkeria capensis, Dorosuchus neoetus, Halazhaisuchus qiaoensis, and Osmolskina czatkowicensis, but this is beyond the scope of the current paper.

Although Halazhaisuchus qiaoensis, ‘Turfanosuchus shageduensis’ and Euparkeria capensis form a clade in our analysis, this result must be considered provisional as only three of the putative euparkeriid taxa were included in the analysis and support for the clade was low, with osteoderm shape constituting the only synapomorphy of the clade (with osteoderms not preserved in ‘Turfanosuchus shageduensis’). The osteoderms of Halazhaisuchus qiaoensis and Euparkeria capensis are indeed very similar, being leaf-shaped and possessing medially offset longitudinal keels. Moreover, the osteoderms are arranged almost identically in the two taxa, forming in each case two paramedian rows slightly out of step with the spine tables below them. However, very similar osteoderms are found in several crown pseudosuchians (e.g., Ticinosuchus ferox—Krebs, 1965; Rauisuchus tiradentes—Lautenschlager, 2008), and doswelliids (e.g., Jaxtasuchus—Schoch & Sues, 2013) show some osteoderms longer than wide, though not of the same shape as those of Halazhaisuchus qiaoensis. Unlike the condition in doswelliids (Desojo, Ezcurra & Schultz, 2011), no characteristics of the osteoderms appear to be unique to Euparkeriidae. Euparkeria capensis also differs from Halazhaisuchus qiaoensis in lacking the pronounced scapular tuber for muscle attachment that is an apparent autapomorphy of the latter taxon, and in more subtle aspects of shape in several elements (e.g., Euparkeria capensis has a less well developed olecranon process of the ulna, and a slightly less strongly expanded distal end of the scapula). It should also be noted that our scoring for osteoderm shape differs from that of Nesbitt, 2011: character 407), who scored the osteoderms of Euparkeria capensis as “square-shaped, about equal dimensions” rather than “longer than wide”. We disagree with this scoring as the maximum width to maximum length ratio of the paramedian osteoderms of Euparkeria capensis is 0.43 (UMZC T.692j). This is similar to the value for Batrachotomus kupferzellensis (0.46, SMNS 90018), which Nesbitt (2011) scored as having elongated osteoderms, but strikingly different from that for Hesperosuchus agilis (0.72, AMNH FR 6758, measured from Fig. 50 in Nesbitt, 2011) which Nesbitt (2011) scored as having square-shaped osteoderms. The width to length ratio is 0.47 in Halazhaisuchus qiaoensis (IVPP V6027-8), again more similar to the condition in Batrachotomus kupferzellensis than that in Hesperosuchus agilis.

Although ‘Turfanosuchus shageduensis’ is not diagnosable, some ecological inferences can be drawn from the specimen based on tooth shape and forelimb/hind limb ratios. The animal can be estimated to have been around 1.5 m in length, based on length estimates for Euparkeria capensis (Ewer, 1965; Remes, 2007; Botha-Brink & Smith, 2011) scaled according to the length ratio between its femur (127 mm) and the longest femur of Euparkeria capensis (78 mm, SAM-PK-10671). ‘Turfanosuchus shageduensis’ was also probably carnivorous, based on the apparently cylindrical shape of the preserved teeth, though no details of dental morphology can be discerned. Locomotor ability similar to that posited for Euparkeria capensis, namely quadrupedal locomotion and possibly facultative bipedality at speed (Ewer, 1965; Santi, 1993), can be tentatively ascribed to ‘Turfanosuchus shageduensis’: the humerus/femur ratio (1.51 for IVPP V6028), femoral length as percentage of femur+tibia length (55% for IVPP V6028), and humerus+ulna length as a percentage of femur+tibia length (69% for IVPP V6028) are similar to the corresponding values for Euparkeria capensis (1.40, approximately 63%, and 67%, respectively; Ewer, 1965; Gauthier et al., 2011). Femoral morphology is also similar, though the tibia is less symmetrical mediolaterally than that of Euparkeria capensis. Lack of preservation of the pelvic girdle precludes further conclusions regarding locomotor ability. Halazhaisuchus qiaoensis does not possess hind limb material or teeth, precluding similar inferences, although the forelimb elements and pectoral girdle are very similar in shape to those of ‘Turfanosuchus shageduensis’. The double row of paramedian osteoderms possessed by Halazhaisuchus qiaoensis may have assisted in locomotion, as has been posited for Euparkeria capensis (Ewer, 1965), and the osteoderms present do not seem extensive enough to have provided significant protection.

Our reassessment of the putative Chinese Euparkeriidae helps to shed light on character evolution leading up to the origin of archosaurs. Together with Euparkeria capensis, the morphology of ‘Turfanosuchus shageduensis’ and of Halazhaisuchus qiaoensis probably approaches that of the common ancestor of Phytosauria+Archosauria. Whilst the locomotor apparatus of Euparkeria capensis and ‘Turfanosuchus shageduensis’ is not specialized for fully upright or bipedal locomotion, unlike that of early dinosauriforms and pseudosuchians (see Gauthier et al., 2011), it departs from that of more sprawling taxa, with reduction and ventral displacement of the fourth trochanter. Based on Halazhaisuchus qiaoensis, ‘Turfanosuchus shageduensis’ and Euparkeria capensis, the ancestor of Archosauria and phytosaurs can also be hypothesised to have been relatively small and gracile, terrestrial, and probably carnivorous. The vertebrae of Halazhaisuchus qiaoensis show some structures that correspond to the extensive laminae and fossae of Archosauria, phytosaurs, and some other non-archosaurian archosauriforms (e.g., erythrosuchids—Gower, 2003; Cuyosuchus—Desojo, Arcucci & Marsicano, 2002). Such features have often, but controversially, been considered to indicate the presence of pneumatic diverticula (see Butler, Barrett & Gower, 2012), but in Halazhaisuchus qiaoensis these structures are not particularly well developed. Halazhaisuchus qiaoensis, along with Euparkeria capensis, is intermediate in development of vertebral laminae and fossae between crown archosaurs and phytosaurs on the one hand and more basal taxa such as Proterosuchus fergusi on the other. However, laminae and fossae are better developed in Erythrosuchus africanus than in Euparkeriidae (Gower, 2003; Butler, Barrett & Gower, 2012) despite the more crownward placement of the latter, implying that the elaboration of laminae and fossae in archosauriform evolution (whether related to pneumaticity or not) did not follow a simple trend.

Supplemental Information

Table S1 Supplementary table of measurements

Measurements holotypes of Halazhaisuchus qiaoensis and ‘Turfanosuchus shageduensis’ and of holotype and paratypes of ‘Wangisuchus tzeyii’.

Click here for additional data file.

Table S2 Supplementary table of Chinese names

Names of localities, towns, administrative divisions and formations used in this article, showing their equivalents in simplified Chinese, Pinyin, and previously published romanizations and translations.

Click here for additional data file.

Matrix S1 Character matrix

Matrix based on that of Butler et al. (2014) with scores for Halazhaisuchus qiaoensis and ‘Turfanosuchus shageduensis’, as well as for a single Halazhaisuchus qiaoensis OTU incorporating ‘Turfanosuchus shageduensis’ as a junior subjective synonym.

Click here for additional data file.

The authors thank AB Heckert, JB Desojo and an anonymous reviewer for their comments and suggestions, which greatly improved the quality of the manuscript. The authors also thank K Tang (University College London) for assistance with Chinese language names.

Institutional Abbreviations

AMNH American Museum of Natural History, New York, USA

BPI Bernard Price Institute for Palaeontological Research, Johannesburg, South Africa

IVPP Institute of Vertebrate Paleontology and Paleoanthropology, Chinese Academy of Sciences, Beijing, China

NHMUK Natural History Museum, London, UK

NM National Museum, Bloemfontein, South Africa

PEFO Petrified Forest National Park, Arizona, USA

PIN Paleontological Institute of the Russian Academy of Sciences, Moscow, Russia

PVL Istituto Miguel Lillo, Tucumán, Argentina

SAM Iziko South African Museum, Cape Town, South Africa

SMNS Staatliches Museum für Naturkunde, Stuttgart, Germany

TTU Texas Tech University Museum, Lubbock, USA

UFRGS Institute of Geosciences, Federal University of Rio Grande do Sul, Porte Alegre, Brazil

UMZC University Museum of Zoology, Cambridge, UK

USNM National Museum of Natural History, Washington, DC, USA

Additional Information and Declarations

Competing Interests

Author Contributions

The authors declare there are no competing interests.

Roland B. Sookias conceived and designed the experiments, performed the experiments, analyzed the data, wrote the paper, prepared figures and/or tables.

Corwin Sullivan and Richard J. Butler conceived and designed the experiments, performed the experiments, analyzed the data, contributed reagents/materials/analysis tools, wrote the paper, prepared figures and/or tables, reviewed drafts of the paper.

Jun Liu conceived and designed the experiments, performed the experiments, analyzed the data, contributed reagents/materials/analysis tools, wrote the paper, reviewed drafts of the paper.

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
