# Peer review of "Systematics of putative euparkeriids (Diapsida: Archosauriformes) from the Triassic of China"

_PeerJ, doi:10.7717/peerj.658_

## Round 0.1 · original submission · Major Revisions

· Academic Editor

Major Revisions

First of all, I would like to apologize for the delay. With the holiday season, it has taken longer than usual to gather reports from the reviewers.

As you will see from their report, the three reviewers agree that your paper is both well written and scientifically sound. However, they have all made valuable suggestions that would significantly improve your manuscript. Please, revise your manuscript based on these suggestions. Reviewers 2 and 3 have notably identified a number of important issues that would need to be addressed before I can accept your manuscript. The following comments should be addressed in priority:

Reviewer 1:
[Note that this reviewer has attached an annotated manuscript, but that pages with edits were scanned in reverse order. Please also disregard any comment from this reviewer regarding the formatting of citations for references with three authors. He informed me that he realized only later on that PeerJ only uses ‘et al.’ for four or more authors.]
— Rearrange the abstract.
— Clarify the situation with the original diagnosis of Halazhaisuchus (see also comment from Reviewer 2).
— Figure 1: ‘pale gray’ indeed looks almost white when printed. Why not make it white and be done with it? (Please, remember to modify the caption if you provide a new version of this figure)
— Captions of Figures 3 and 5: Please, address the issues enlightened by the reviewer. Be aware that the captions will appear as they are entered online in the manuscript submission system (BTW, there is no need to include them in your manuscript file). This is where you should proof them.

Reviewer 2:
[Note that this reviewer has attached an annotated manuscript]
— Make your diagnosis of Halazhaisuchus qiaoensis more differential.
— Provide more evidence to support the synonymy of ’T.’ shageduensis with H. qiaoensis.
— Include more comparisons in your description of Halazhaisuchus.
— Provide more information regarding the robustness of your euparkeriid clade (it is important as you claim this is the first time euparkeriid are found to be monophyletic).
— You might also want to reconsider Figure 1 based on the comments from the three reviewers, notably Reviewers 2 and 3 who ask for more detailed information regarding the localities, provinces, etc.

Reviewer 3:
[Note that this reviewer has attached an annotated manuscript. Disregard any comments from this reviewer concerning the tables, which she has not been able to find on the system.]
— Improve the descriptions of some skeletal elements (see annotated pdf).
— Include more comparisons with other archosauriformes in your description.

Additional comments from the Editor:
I have made a number of comments on an annotated pdf, most of which concern formatting. I will send it to you in a separate email because the system does not offer me the possibility of attaching a file to this decision letter. There are also several issues with missing references, or conversely with references appearing in the reference list but not cited in the text. Please be sure to address all these as well in your revised manuscript.

·

Basic reporting

Reporting in this article is basically sound. I have some quibbles with the order of presentation. Namely, Wangisuchus tzeyii is discussed and illustrated first in the paper, but last in the abstract. There are also a variety of minor comments on the English (basic wordsmithing needed, as is typical of most manuscripts). As an example of the latter, the authors consistently use "around" when they mean "approximately" and a few other minor edits. They also use "date" when they mean "age" for strata.

There is also some confusion regarding figure call-outs during the description of cervical vertebrae (lines 396-402 or so). More precision is desired here.

The authors do need to make clear what the nature of the "original diagnosis" is for Halazhaisuchus---was this translated? Is it verbatim? Precise citation (incl. page numbers) is desired.

Regarding the figures:
Figure 1. What is called "light gray" in the caption sure looks white to me on my printed copy.

Figure 3. Italicize 'Turfanosuchus' shageduensis; note also that text and figure have "pra" but caption has "pr" for prearticular

Figure 5. Italicize Batrachotomus kupferzellensis in caption.

Experimental design

This is a standard paleontological paper, consisting of modern redescription and reanalysis of previously described material. The methods and techniques are competently executed.

Validity of the findings

The findings appear reasonable (there's no way I care to replicate the phylogenetic analysis). I agree with the general taxonomic assessments of the authors. On the attached manuscript I have made some notes where they could be more clear. In particular, some of the material is so poorly preserved that, although they are reasonable to consider it congeneric with the better preserved material, it is not clear how they can be "certain" of morphology that is not preserved.

Additional comments

A generally sound manuscript that should require only minor editing to be acceptable. My biggest problems are with the abstract (which should be reorganized to match the layout of the paper, and some passive voice re-written). Otherwise, I think this is a viable manuscript.

Reviewer 2 ·

Basic reporting

The manuscript satisfies all the areas of the basic reporting guidelines

Experimental design

The overall design of the study and analyses are well-constructed; I have no major comments here. The research question is well-defined and thought-out, and the authors do a good job of answering that question.

Validity of the findings

I have one principle criticism of the manuscript, which is the synonymy of 'T.' shageduensis (IVPP V6028) with Halazhaisuchus qiaoensis (IVPP V6027). Saying that these two specimens are found as sister taxa in a phylogenetic analysis and that they are "nearly identical" is not sufficient for a formal referral. Alpha taxonomy is largely independent of phylogeny (i.e., two valid and separate taxa can be sister taxa, just the same as two specimens of a single taxon can be recovered as sister taxa); you need to demonstrate that Halazhaisuchus has a valid diagnosis (which you've done), and that enough character states are preserved in 'T.' shageduensis to satisfy the diagnosis of Halazhaisuchus (which you have not done). No where in the manuscript do you unambiguously discuss comprehensively which features of the diagnosis are present in 'T.' shageduensis to justify your referral. Furthermore, of the two autapomorphies that diagnose Halazhaisuchus, one is not preserved in 'T.' shageduensis, and the other is absent. This severely weakens your proposed synonymy. At this point, you've only demonstrated that 'T.' shageduensis and Halazhaisuchus are both euparkeriid archosauriforms. Based on the evidence presented, its very possibly that 'T.' shageduensis is an indeterminate euparkeriid archosauriform rather than a specimen of Halazhaisuchus. Therefore, you need to do a better job discussing how 'T.' shageduensis satisfies the diagnosis of Halazhaisuchus.

Additional comments

Overall, this is a very nice redescription of some long-neglected material. It certainly deserves publication in PeerJ after revision. I have a couple of other significant comments, which are listed below. Other comments and edits are attached as an annotated PDF of the manuscript using the comment and mark-up functions of Adobe Acrobat.

- The 'unique combination of character states' portion of the diagnosis needs to be made differential. That is - which characters are present in other taxa. This is usually presented in the form of: "differs from all other taxa except X, Y, and Z in the presence of character 1; shares with all early archosauriforms except taxon A the absence of character 2; etc"

- The original diagnosis is not particularly informative - I think you can delete it to save space.

- The description is largely devoid of comparisons to other taxa. In modern paleontology, comparative descriptions are the standard - when you describe morphology in isolation it is of little use without a comparative context. Even comparisons to Euparkeria are sparse. Throughout the description you need to say how the features present in Halazhaisuchus are similar or different from Erythrosuchus, Vancleavea, proterochampsids, doswelliids, Euparkeria, Dorosuchus, and phytosaurs.

- I find Figure 1 in its present state to be of limited value, because it is at such a large scale. Please reduce the map of all China to be smaller, and then include a larger inset map of just the region where the specimens are found, which includes more geographic detail.

- please add a close-up photo(s) of the osteoderms in dorsal view

Annotated reviews are not available for download in order to protect the identity of reviewers who chose to remain anonymous.

·

Basic reporting

Overall I found this to be a well-written description of interesting specimens with some important implications for euparkeriid taxonomy, specifically a departure from more than five decades of convention of the monophyletic family Euparkeriidae. I think that based on the evidence provided the Halazhaisuchus is almost certainly novel and that this redescripction will change some of the ways we conduct archosauriforms taxonomy. I think that this manuscript is acceptable for publication but with some minor revisions. The majority of my changes are on the attached edited copy of the manuscript; however, I would like to emphasise a few off these suggestions below.

Experimental design

ok. I can not find the Table S1, S2, and S3. S1 and S3 are very important and should be check in detail.

Validity of the findings

The conclusions are acceptable, but they should compare with more archosauriform taxa in the description (e.g. one proterochampsid, one doswelliid, one phytosaur), and enrich the description of some elements (e.g. osteoderms, vertebrae, coracoid).

Additional comments

Please, see attach comments on the pdf. Pay attention to the editorial rules, figure indications, references, text citations, include the Table files, boostrap values. Congrats for the paper.

---

## Round 0.2 · Minor Revisions

· Academic Editor

Minor Revisions

You have done a great job incorporating the reviewers comments and restructuring your manuscript. One of the initial reviewers confirmed that you have dealt with all his previous concerns. However, I have spotted some minor problems that need to be addressed before the manuscript can go into production. Given the amount of changes between this version and the first one, it is totally understandable that some errors have crept into the text. You should be able to deal with my minor requests fairly quickly.

- Abstract: italicize the taxon names in the abstract (the version entered online in the submission system, which will be used for production)

- Throughout the manuscript: homogenize the use of quotes for Turfanosuchus shageduensis (around genus only, or around genus+species)

- Lines 324, 358, 386, 470, 489, 507, 509, 511, 519, 662, 666, 683, 688, 726, 779, 837, 841, 893, 927, 945, 946, 948, 953, 954: MCZ, NMQR, USNM, NHMUK, PEFO, PIN, UFRGS, PVL, BPI, TTU, NM are missing in the ‘Institutional Abbreviations’ section.

- Lines 530 and 739: ‘SAM-Pk-5867’ should be ‘SAM-PK-5867’

- Line 683: ‘in press’ should not be in italics

- Lines 747–748 and 1060–1062: (Nesbitt, 2011) is cited three times in the same sentence, one time should suffice

- Line 869: complete citation ‘(Gower)’

- Line 926: complete citation ‘Lecuona & Desojo’

- Line 1050: remove hyphen before ‘Euparkeria’

- Line 1181: the reference Dilkes and Sues (2009) is missing

- Lines 1186–1188: the reference Gower and Sennikov is duplicated (correctly placed in line 1232)

- Lines 1239–1240: reorder Gow (1975) alphabetically

- Line 1422: the reference Spielmann et al. (2008) is missing

- Caption of Fig. 8: ‘pa’ should be ‘pra’ (please updated the caption in the online submission system)

Reviewer 2 ·

Basic reporting

I previously reviewed an earlier version of this MS for PeerJ. All of my concerns have now been dealt with and I consider the manuscript ready for acceptance and publication.

Experimental design

see above

Validity of the findings

see above

Additional comments

I previously reviewed an earlier version of this MS for PeerJ. All of my concerns have now been dealt with and I consider the manuscript ready for acceptance and publication.

---

## Round 0.3 · accepted · Accept

· Academic Editor

Accept

Thank you for these last edits. I can now formally accept your manuscript.

One last thing, you forgot to add NMQR in the Institutional Abbreviations. You will be able to add this at proof stage though.